

# Network connectivity between the winter Arctic Oscillation and summer sea ice in CMIP6 models and observations

William Gregory[1], Julienne Stroeve[1,2,3], and Michel Tsamados[1]

[1]Centre for Polar Observation and Modelling, Earth Sciences, University College London, UK
[2]National Snow and Ice Data Center, Cooperative Institute for Research in Environmental Sciences, University of Colorado, Boulder, Colorado, USA
[3]Centre for Earth Observation Science, University of Manitoba, Winnipeg, MB, Canada

**Correspondence:** William Gregory (william.gregory.17@ucl.ac.uk)

**Abstract.** The indirect effect of winter Arctic Oscillation (AO) events on the proceeding summer Arctic sea ice extent suggests an inherent winter-to-summer mechanism for sea ice predictability. On the other hand, operational regional summer sea ice forecasts in a large number of coupled climate models show a considerable drop in predictive skill for forecasts initialised prior to the date of melt onset in spring, suggesting that some drivers of sea ice variability on longer time scales may not

be well represented in these models. To this end, we introduce an unsupervised learning approach based on cluster analysis and complex networks to establish how well the latest generation of coupled climate models participating in phase 6 of the World Climate Research Programme Coupled Model Intercomparison Project (CMIP6) are able to reflect the spatio-temporal patterns of variability in northern-hemisphere winter sea-level pressure and Arctic summer sea ice concentration over the period 1979–2020, relative to ERA5 atmospheric reanalysis and satellite-derived sea ice observations respectively. Two specific global

metrics are introduced as ways to compare patterns of variability between models and observations/reanalysis: the Adjusted Rand Index – a method for comparing spatial patterns of variability, and a network distance metric – a method for comparing the degree of connectivity between two geographic regions. We find that CMIP6 models generally reflect the spatial pattern of variability of the AO relatively well, although over-estimate the magnitude of sea-level pressure variability over the north-western Pacific Ocean, and under-estimate the variability over the north Africa and southern Europe. They also under-estimate

the importance of regions such as the Beaufort, East Siberian and Laptev seas in explaining pan-Arctic summer sea ice area variability, which we hypothesise is due to regional biases in sea ice thickness. Finally, observations show that historically, winter AO events (negatively) covary strongly with summer sea ice concentration in the eastern Pacific sector of the Arctic, although now under a thinning ice regime, both the eastern and western Pacific sectors exhibit similar behaviour. CMIP6 models however do not show this transition on average, which may hinder their ability to make skilful seasonal to inter-annual

predictions of summer sea ice.



# 1 Introduction

Arctic sea ice is a key component of the polar climate system, acting as a barrier which both reflects incoming solar radiation and regulates the rate of energy exchange between the atmosphere and ocean. Over the past four decades however, it can be seen as a direct barometer for climate change, having suffered significant losses in areal extent across all seasons (Stroeve and Notz, 2018), linked to the long-term increase in anthropogenic $CO_2$ emissions (Notz and Stroeve, 2016). Such sea ice decline has profound implications for regional northern-hemisphere circulation patterns (Francis et al. 2009; Cohen et al. 2014, 2020), ecological productivity (Sakshaug et al. 1994; Stirling 1997; Stroeve et al. 2021), and coastal communities (Fritz et al. 2017; Larsen et al. 2021) in present and future decades, as many model studies now predict the increasing likelihood of a seasonally ice-free Arctic Ocean occurring before the middle of this century (Stroeve et al. 2007; Jahn 2018; Notz and SIMIP-Community 2020; Årthun et al. 2021).

Imprinted on this observed sea ice decline is a pattern of significant inter-annual variability, particularly in the summer months (e.g., Onarheim et al. 2018; Stroeve and Notz 2018). Subsequently, understanding the drivers of such variability has important consequences for our ability to make reliable sea ice predictions on seasonal to inter-annual time scales. A number of studies for example have highlighted various climatological teleconnections as key drivers of sea ice variability, including land-ice interactions (typically via the atmosphere; Serreze et al. 1995; Overland et al. 2012; Matsumura et al. 2014; Crawford et al. 2018), atmosphere-ice (Deser et al. 2000; Kapsch et al. 2013; Park et al. 2018; Olonscheck et al. 2019), ocean-ice (Venegas and Mysak 2000; Vinje 2001; Zhang 2015), and ice-ice (Schröder et al. 2014; Bushuk and Giannakis 2017), suggesting inherent sources of sea ice predictability across the various components of the climate system. Furthermore, coupled climate models have demonstrated a horizon of sea ice predictability beyond 12 months lead time based on so-called 'perfect-model' experiments (Holland et al. 2011; Tietsche et al. 2014; Day et al. 2014; Bushuk et al. 2019). Operationally however, regional summer sea ice forecasts (in models) appear to be strongly controlled by a 'spring predictability barrier' (Bonan et al., 2019), which is governed by the date of melt onset in the preceding spring (Bushuk et al., 2020). This leads to the question as to how well climate models reflect the teleconnections known to drive summer sea ice variability, given the gap between perfect-model and operational regional forecast skill in those models (Bushuk et al., 2019). Recent work has gone into investigating physically-based mechanisms for sea ice predictability (Bushuk and Giannakis 2017; Bonan and Blanchard-Wrigglesworth 2020; Giesse et al. 2021) in order to assess whether the shortfalls in operational climate model forecasts can, in part, be attributed to lack of representation of such mechanisms across a wide range of general circulation models (GCMs).

In this study we pursue a similar line of investigation, looking specifically at the Arctic Oscillation (AO) teleconnection (Thompson and Wallace, 1998), whose winter pattern has been shown to explain up to 22% of the variability in pan-Arctic September sea ice extent (Park et al., 2018). Historically, the dominant spatio-temporal modes of winter sea-level pressure variability have been somewhat misrepresented in the majority of GCMs participating in previous phases of the Coupled Model Intercomparison Project (CMIP), e.g., CMIP3 (Miller et al. 2006; Cattiaux and Cassou 2013) and CMIP5 (Jin-Qing et al. 2013; Gong et al. 2016), which naturally has implications for the representation of the AO to sea ice teleconnection in those models. Here, we assess the spatio-temporal patterns of variability in both winter sea-level pressure and summer sea ice concentration,





and also the presence of the winter AO to summer sea ice teleconnection over the period 1979–2020, in 31 of the latest generation of GCMs submitted to CMIP6 (Eyring et al., 2016), and how this teleconnection may be changing over time as the ice cover thins and is more susceptible to atmospheric forcing (Maslanik et al. 1996; Mioduszewski et al. 2019).

Similar to previous works (Fountalis et al. 2014, 2015; Gregory et al. 2020), our method here is based on complex networks; an approach which provides a relatively simple and visual framework with which to analyse and display large volumes of data that typically represent complex physical systems. Their use across multiple disciplines has grown considerably over recent decades, with intuitive applications in computer science and social networks (Albert and Barabási 2002; Newman 2003; Boccaletti et al. 2006; Cohen and Havlin 2010) – known as *structural networks*, to more abstract applications in e.g., neuroscience (Zhou et al. 2007; Morabito et al. 2015; delEtoile and Adeli 2017), seismology (Abe and Suzuki, 2006), and climate science (Tsonis and Roebber 2004; Tsonis et al. 2006; Donges et al. 2009; Fountalis et al. 2014; Dijkstra et al. 2019; Gregory et al. 2020) – known as *functional networks*. Climate network analysis was first introduced by Tsonis and Roebber (2004), and has subsequently proven to be a powerful tool kit for extracting statistical information from the array of climatological teleconnections that govern climate variability, and is a useful addition to more conventional approaches of analysing spatio-temporal patterns of variability, such as empirical orthogonal function analysis (Donges et al., 2015). This consistency allows us to use the network framework to derive climatological teleconnections which are traditionally defined as the leading mode of variability of their respective climate fields.

This paper is structured as follows: in Sect. 2 we introduce the sea ice observations, atmospheric reanalysis, and CMIP6 model data that are used to generate complex networks of the respective sea ice and atmospheric fields. In Sect. 3 we introduce the complex networks methodology as well as two global metrics which are used to describe similarities and differences between networks. In Sect. 4 we present the results of the networks generated from the 31 different CMIP6 model outputs and discuss their similarities and differences relative to the observations and reanalysis data, and furthermore evaluate the presence of the winter AO to summer sea ice teleconnection across all models. We then end with a discussion and conclusions in Sect. 5.

## 2 Data

### 2.1 Observations

For analysing summer sea ice concentration variability in the observations, we compute the average of monthly mean June, July, August and September (JJAS) sea ice concentration fields between 1979 and 2020 from three separate observational data sets based on the series of multi-frequency passive microwave satellite observations since October 1978. These include the National Snow and Ice Data Center (NSIDC) NASA Team (Cavalieri et al., 1996), and Bootstrap (Comiso, 2017) products, as well as the Ocean and Sea Ice Satellite Application Facility (OSI-SAF) OSI-450 (1979–2015) and OSI-430-b (2016–2020) products (OSI-SAF 2017; Lavergne et al. 2019). We use three different products as each has subtle variations in their summer variability, from how each account for new melt-pond formation (Comiso et al., 2017). Each of the three data sets apply separate processing algorithms to passive microwave brightness temperatures derived from multiple satellites across the historical record: Nimbus-7 SMMR (1979–1987), the DMSP F-8, F-11 and F-13 SSM/Is (1987–2007), and finally the DMSP F-18 SSM/I (2007–2020).





These data are provided on $25 \times 25$ km polar stereographic (NASA Team and Bootstrap), and EASE (OSI-450 and OSI-430-b)
grids, which are re-gridded to a common $50 \times 50$ km polar stereographic grid using a nearest neighbour interpolation here for
computational reasons. Grid cell area information (used to generate area-weighted time series, see Sect. 3.1) was also extracted
from NSIDC's pixel area tools library.

## 2.2  Atmospheric Reanalysis

The AO is typically defined as the leading mode of variability in mean sea-level pressure data north of $20°$ N (Thompson
and Wallace, 1998). As a proxy for an observational record of the winter AO here, we compute the average of monthly mean
December, January, February and March (DJFM) mean sea-level pressure data north of $20°$ N, from ERA5 reanalysis (ERA5,
2017). We only use one reanalysis product due to the high consistency of sea-level pressure fields between different reanalyses
over the Arctic region (Graham et al., 2019). As December data are not available for the year 1978 for ERA5, the winter period
in 1979 corresponds to the average of January, February and March data. Sea-level pressure fields are output on a $2° \times 4°$
latitude-longitude grid.

In Sect. 5 we also make use of the Pan-Arctic Ice Ocean Modeling and Assimilation System (PIOMAS) sea ice thickness model
(Zhang and Rothrock, 2003) to help explain some of the features related to the winter AO to summer sea ice teleconnection.
PIOMAS is a coupled ice-ocean model that assimilates observed sea ice concentration and sea surface temperatures (open
water only), and is forced by NCEP-NCAR atmospheric reanalysis. Although it is a model, it has been shown to be relatively
consistent with in-situ and submarine observations (Schweiger et al., 2011), and generally has consistent biases with CMIP3/5
models relative to observational data, in terms of its ice thickness distribution (Stroeve et al., 2014). Furthermore, it is able to
provide consistent coverage over the observational period.

## 2.3  CMIP6 model outputs

We assess the seasonal patterns of variability in sea ice concentration, and mean sea-level pressure outputs from 31 different
GCMs participating in CMIP6. In order to compare with recent observations, we combine monthly averaged model outputs
from historical runs (1979–2014) with ScenarioMIP run SSP5-8.5 (Gidden et al., 2019) to extend the analysis period to 2020,
hence only model ensembles which contain historical and ScenarioMIP outputs for both sea ice concentration and mean sea-
level pressure are considered in this work. As detailed above, we compute the corresponding winter (DJFM) and summer
(JJAS) averages for mean sea-level pressure and sea ice concentration outputs respectively. Sea ice concentration outputs are
also re-gridded to the same $50 \times 50$ km polar stereographic grid as the observational data sets, and mean sea-level pressure
outputs are re-gridded to a $2° \times 4°$ latitude-longitude grid. The chosen models, along with their respective number of available
ensembles are summarised in Table 1.





## 3  Method

### 3.1  Complex networks

Generating complex networks here follows the methodology from previous studies (Fountalis et al. 2014; Gregory et al. 2020). In this section we summarise the key steps, and present an example DJFM sea-level pressure network from ERA5, as well as JJAS sea ice concentration networks from the observations.

In general, we can consider a network as a group of vertices, or *nodes*, whereby each node $k$ may be connected to any
other node in the network $l$ via a weighted edge, or *link*. Subsequently, in our implementation a climate network of $N$ nodes corresponds to time series data $\boldsymbol{G} = \{\boldsymbol{g}_k\}_{k=1}^N$ representing $n$ regularly sampled observations in time $\boldsymbol{g}_k = (g_{1k}, g_{2k}, \ldots, g_{nk})$ at $N$ fixed geographical locations, and the links represent statistical interdependencies between any pair of node time series $\boldsymbol{g}_k$ and $\boldsymbol{g}_l$. In more detail, let us define $\mathbf{X} = \{\boldsymbol{x}_p\}_{p=1}^P$ as a de-trended (zero-mean) time series data set (e.g., DJFM mean sea-level pressure anomalies or JJAS sea ice concentration anomalies), which represents $n$ regularly sampled observations in time
$\boldsymbol{x}_p = (x_{1p}, x_{2p}, \ldots, x_{np})$ at $P$ fixed geographical locations, such that $\mathbf{X} \in \mathbb{R}^{P \times n}$. The $N$ network nodes are then derived by implementing a grid-based clustering algorithm to the input data set, so that the dimensionality of $\mathbf{X}$ is reduced from $P$ to $N$ (see Gregory et al. 2020 for further details of this clustering step). Each cluster (node) then corresponds to a particular pattern of climate variability and represents a spatial region of e.g., summer sea ice concentration or winter sea-level pressure that has behaved in a homogeneous way over the length of the time series record. We can then generate links between the nodes by first
computing the cumulative anomaly time series of each network node, which for a given node $\mathcal{C}_k$, is taken as the sum of the grid-cell-weighted de-trended time series of all cells within that node:

$$\boldsymbol{g}_k = \sum_{p \in \mathcal{C}_k} \mathbf{X}_p \sqrt{\mathrm{w}_p}, \tag{1}$$

where $\mathrm{w}_p = \cos(\theta_p)$ for a regular latitude-longitude grid ($\theta_p$ is the latitude of grid cell $p$), or simply $\mathrm{w}_p = d_p$ for a polar stereographic area grid ($d_p$ is the area in km$^2$ of grid cell $p$). Subsequently, the link weight between two nodes $k$ and $l$ is
calculated as the temporal covariance between two network node anomaly time series:

$$w_{kl} = \frac{1}{n-1} \sum_{i=1}^n (g_{ik} - \mathbb{E}[\boldsymbol{g}_k])(g_{il} - \mathbb{E}[\boldsymbol{g}_l]). \tag{2}$$

Finally, we define the weighted degree, or *strength*, of a given network node $\mathcal{S}_k$ as the sum of the absolute value of all its associated link weights:

$$\mathcal{S}_k = \sum_{l=1}^N |w_{kl}|. \tag{3}$$

The nodes with the highest strength are commonly referred to as the *hubs* of the network (Tsonis and Roebber, 2004), and represent the dominant patterns of variability of the input data set $\mathbf{X}$. By this definition, the node with the highest strength





belonging to the network of mean sea-level pressure data can be considered a good proxy for the AO. Note also that this network framework allows for weighted links between nodes of a single network (as detailed above), and also between nodes of multiple networks (i.e., between nodes of sea-level pressure and sea ice concentration), which is used to assess the winter AO
to summer sea ice teleconnection in Sect. 4.3.

Fig. 1 shows the network structure of DJFM mean sea-level pressure data from ERA5. In this case we can see how the network nodes correspond to a set of spatially contiguous areas, where for a given node, each grid cell is weighted by the strength of the node in which that cell belongs. The node with the highest strength in Fig. 1 can be considered as a proxy for the spatial
pattern of variability of the AO, and from this map we can see that this corresponds to the large node situated over the majority of the Arctic Ocean, Greenland, the Canadian Archipelago, and parts of northern Russia. The weighted links then illustrate how each of the nodes have covaried relative to each other over the period 1979–2020, and indeed we notice the out-of-phase relationship (negative covariance) between the AO node and the mid-latitude Atlantic sector; highlighting the dipole nature of the North Atlantic Oscillation (Hurrell et al., 2003).
We can also extract the temporal component of variability from the 'AO node' (Fig. 2), which produces a very consistent signal with the standard AO index as defined by the National Oceanic and Atmospheric Administration (NOAA), highlighting the robustness of the complex networks method. It is worth noting however that in Fig. 2, and indeed for the rest of this manuscript, we reverse the sign of the temporal component of each node of the winter sea-level pressure networks (from ERA5 and CMIP6 data) in order to be consistent with the standard AO index, for which positive AO index values correspond to low atmospheric
pressure, and similarly negative AO index values correspond to high atmospheric pressure.

In Fig. 3 we show similar networks for JJAS sea ice concentration, from each of the observational products. Here we can see how the dominant patterns of summer sea ice variability (i.e., highest node strengths) are typically in the East Siberian and Laptev seas, as well as the Canada basin. Each observational product generally shows the same structure of largely positive
covariance between network nodes, and the out-of-phase connection between the Fram Strait and the Pacific sector. The magnitude of the node strengths between the observational data sets somewhat varies in the dominant regions of variability, with the Bootstrap product showing the largest strengths in the East Siberian and Laptev seas. In the next section we introduce two global metrics for deriving quantitative measures of similarity between networks.

### 3.2   Metrics for comparing networks

Before we introduce the two metrics which are used to compare similarities between complex networks, it is worth saying a few words about what information we can expect to obtain when comparing models and observations/reanalysis. Due to the fact that any CMIP6 model ensemble is in its own phase of internal variability (e.g., Hawkins and Sutton 2009; Notz 2015), we cannot expect to find consistency in the sign and magnitude of anomalies between e.g., the ERA5 AO time series, and that of any one model ensemble (and similarly for sea ice), therefore it would not be prudent to perform any analysis which makes
direct comparisons of any time series between observational and CMIP6 network nodes. We can however expect a model with





accurate physics to reproduce similar dominant regions of variability as the observations, and the same sign and magnitude of the inter-connected links between nodes, e.g., the strong negative coupling between the AO and sea-level pressure anomalies in the north Atlantic, and the weak negative coupling with the north Pacific (see Fig. 1), and similarly we can expect the same regional responses of Arctic sea ice to different phases of the AO between observations and models. The two metrics

we introduce in the coming sections provide a way to quantify similarities in the locations which the observations and models define as the dominant regions of variability, and the connectivity of these regions, without explicitly comparing network node time series.

### 3.2.1    Adjusted Rand Index

The Adjusted Rand Index (ARI; Hubert and Arabie 1985) is a metric which is often used to evaluate similarities between sets

of clusters (Steinley, 2004), and as such we use it here to compare how two networks have clustered grid cells together to form their spatially contiguous set of network nodes, and subsequently their spatial patterns of either sea-level pressure or sea ice concentration variability. To understand the ARI, it is worth briefly introducing the (un-adjusted) Rand Index by following the example outlined by Rand (1971). First, consider two different synthetic networks which have clustered grid cells together in two distinct ways:

$\text{Network}_1 = [(a, b, c), (d, e), (f)]$

$\text{Network}_2 = [(a, b), (c, d), (e, f)].$

In these two simple network constructions, there are 3 nodes in each network, where each node contains a clustering of cells labelled $a - f$. The Rand Index then measures similarities and differences in the clustering of these cells by analysing all the possible cell pairings between the two networks (see Table 2). In this example there are a total of 10 similarities (grid cells

which are clustered together in both networks *and* grid cells which are separate in both networks) out of a possible 15 pairings, which gives a Rand Index score of 10/15 = 0.67. The ARI is then an update of the Rand Index, which takes into account the fact that grid cells could be clustered together by chance (see Hubert and Arabie 1985 for further details). In the synthetic example above, the ARI corresponds to 0.07 – note that the ARI varies between 0 (totally dissimilar clustering) and 1 (identical clustering). Computing the ARI between e.g., the NASA Team and OSI-SAF summer sea ice concentration networks produces

a value of 0.69, showing relatively consistent clustering (as we saw qualitatively in Fig. 3).

### 3.2.2    Network distance metric

The network distance metric ($\mathcal{D}$; Fountalis et al. 2015) provides a way to compare networks in terms of both the spatial extent of their network nodes, and also their node strengths. Recall that node strength incorporates information about the connectivity of a particular node (i.e., the magnitude of all its connected links), hence when comparing the strength of a particular region

between models and observations, we can infer which one has a larger degree of covariability across the network. This allows us to deduce whether the models over- or under-estimate the magnitude of variability of a particular region without comparing node time series. Consider for example the underlying map of node strengths in Fig. 3. We can compute $\mathcal{D}$ by first taking the





sum of the absolute difference between two of these 'strength maps' $M_1$ and $M_2$ (e.g., NASA Team and OSI-SAF), and then normalising by the sum of the absolute difference between random permutations of both network strength maps, $\hat{M}_1$ and $\hat{M}_2$:

$$\mathcal{D} = 1 - \frac{\sum_{p=1}^{P} |M_{1p} - M_{2p}|}{\sum_{p=1}^{P} |\hat{M}_{1p} - \hat{M}_{2p}|}. \tag{4}$$


A value of $\mathcal{D} = 1$ means that both networks are identical in their node strengths and spatial extent of nodes, whereas a value close to $\mathcal{D} = 0$ implies that the two networks are as similar as a random assignment of node strengths to grid cells. Computing $\mathcal{D}$ between the NASA Team and OSI-SAF summer sea ice concentration networks produces a value of 0.86.

The combination of ARI and $\mathcal{D}$ allows us to infer various properties between two networks. For example, when ARI $= 1$ and $\mathcal{D} = 0$, this suggests that two networks agree in terms of which grid cells have behaved homogeneously over the length of the time series record in order to cluster together to form network nodes, however they disagree in terms of the magnitude of variability of the nodes. On the other hand, if ARI is close to 0 and $\mathcal{D}$ is close to 1, then this implies that the magnitude of variability across the networks are relatively consistent, however the geographic areas which are clustered together to form network nodes are considerably different. Two networks can then be considered identical if ARI and $\mathcal{D} = 1$.


## 4 Results


### 4.1 Sea-level pressure networks in CMIP6

For every available ensemble from each of the CMIP6 models outlined in Table 1, we compute individual complex networks of DJFM sea-level pressure over the period 1979–2020, and then compute ARI and $\mathcal{D}$ metrics relative to the ERA5 sea-level pressure network. In Fig. 4 we can see how the spread in $\mathcal{D}$ values across all model ensembles is over twice as large as for the ARI values, which suggests large inter-model disagreement on the degree of connectivity of network nodes, and hence the magnitude of regional sea-level pressure variability. Note that the apparent linear relationship between ARI and $\mathcal{D}$ is to be expected, given that both metrics encapsulate information related to the spatial agreement of network nodes between models and ERA5. Across all models, CNRM-CM6-1 (r6i1p1f2) produces the most similar network structure to ERA5, with ARI $= 0.76$ and $\mathcal{D} = 0.80$. Fig. 5 shows the corresponding network for CNRM-CM6-1, and also MIROC-ES2L (r1i1p1f2). The MIROC-ES2L model produces the most dissimilar network structure relative to ERA5, with ARI and $\mathcal{D}$ of 0.50 and 0.04 respectively. The networks show how the CNRM-CM6-1 model produces the relatively consistent node of high strength over the Arctic Ocean (similar to ERA5 in Fig. 1), and also shows the same strong negative linkage with the mid-latitude Atlantic sector, and weak linkage with the Pacific sector. On the other hand, the MIROC-ES2L model shows significantly different regions of variability than ERA5, and also weaker connectivity, with overall weaker link weights and very low strength over the Arctic Ocean. In Fig. 6a-b we average each of the network strength maps across all of the CMIP6 model ensemble members, and compare this with the ERA5 strength map. While this removes the ability to identify individual network nodes and their links, it does allow us to qualitatively assess how CMIP6 models, on average, represent the spatial patterns of winter sea-level pressure variability and their degree of connectivity. We notice for example, that on average CMIP6 models represent the spatial






pattern of the AO relatively well, although slightly under-estimate its node strength. Furthermore, node strengths in the north-
western Pacific Ocean appear to be over-estimated on average, while they are under-estimated over north Africa, and southern
Europe. In Fig. 6c-d we also show the percentage of variance in mean northern-hemisphere sea-level pressure anomalies that
is explained by each ERA5 network node, and the average of CMIP6 nodes (the mean sea-level pressure anomalies in CMIP6
models are computed for each individual model ensemble, and then the percentage of variance is computed between this signal
and its own respective sea-level pressure network nodes). We can see that the nodes centred over the Arctic Ocean explain
the highest percentage of variance in northern-hemisphere sea level pressure in both ERA5 and CMIP6 networks, however the
models under-estimate the relative importance of the north Atlantic and north Africa–southern Europe region in explaining
winter sea-level pressure variability. It is also worth mentioning that although the CMIP6 models identify the north Pacific as
a region of strong covariability (Fig. 6b), the percentage of variance explained by this region is relatively low. This can occur
due to the fact that network nodes which are larger in spatial extent will naturally show higher covariance with other regions
(and hence node strengths), because the temporal component of variability of a given node corresponds to the sum of all grid
cell time series within that node (see Eq. 1). If however a node's correlation (i.e., standardised covariance) with the mean
sea-level pressure signal is relatively small, then this results in a squared reduction in percentage of variance explained (recall,
percentage of variance explained $=$ correlation$^2$). This therefore suggests that sea-level pressure over the north Pacific Ocean
in CMIP6 models is more spatially homogeneous than ERA5.

## 4.2 Sea ice concentration networks in CMIP6

We now compute individual complex networks of JJAS sea ice concentration over the period 1979–2020 for every available
ensemble from each of the CMIP6 models, and then compute ARI and $\mathcal{D}$ metrics relative to the NASA Team, Bootstrap and
OSI-SAF sea ice concentration networks. In Fig. 7 we see a lower spread in ARI values than compared to the $\mathcal{D}$ values which,
similar to the sea-level pressure networks, suggests that the models show large disagreement on the degree of connectivity
of network nodes, and hence the magnitude of regional sea ice concentration variability. What is perhaps noticeable is that
the models which appear to perform better in terms of their summer sea ice ARI and $\mathcal{D}$ scores, are not necessarily the same
as those that score well for their winter sea-level pressure networks – discussed further in Sect. 5. In Fig. 8 we show sea
ice concentration networks from the MIROC6 (r1i1p1f1) and the CAMS-CSM1-0 (r1i1p1f1) models. The MIROC6 model
produces closer patterns of variability to the observations than other CMIP6 models, with ARI values of 0.48 (NASA Team),
0.48 (Bootstrap), 0.47 (OSI-SAF), and $\mathcal{D}$ values of 0.66 (relative to each observational network). Having said that, we can see
that the spatial extent and strength of the node in the Beaufort Sea–Canada basin is somewhat under-estimated (relative to the
observational networks shown in Fig. 3), and the node strength in the Laptev Sea is over-estimated, and interestingly its link
between the Beaufort Sea and the East Siberian Sea is negative. It does however produce consistent out-of-phase network links
between the Fram strait and the Eurasian-Pacific sectors of the Arctic. The CAMS-CSM1-0 model produces a more dissimilar
score with ARI values of 0.33 (NASA Team), 0.30 (Bootstrap), 0.33 (OSI-SAF), and $\mathcal{D}$ values of 0.28 (NASA Team), 0.30
(Bootstrap), and 0.29 (OSI-SAF). The low $\mathcal{D}$ values are being caused by the significant over-estimation in the link weights, and
hence node strengths, in the Greenland, Iceland, and Norwegian Seas, Barents Sea, East Siberian Sea and Laptev Sea (notice





how the link weights and node strengths in this model are in some cases an order of magnitude higher than the observational networks). In Fig. 9a-b we average each of the network strength maps across both the observational data, and across all of the

CMIP6 model ensemble members. Here we notice that, on average, the models show the dominant regions of variability are in the East Siberian and Laptev seas, although the node strengths are somewhat over-estimated relative to the observations. Furthermore, while the observations outline the Beaufort Sea–Canada basin as the region of highest connectivity (more so than the East Siberian–Laptev seas), the models show relatively little connectivity here on average. In Fig. 9c-d we also show the percentage of variance in pan-Arctic summer sea ice area that is explained by each observational network (averaged), and the

average of CMIP6 model ensembles. In this case the models generally under-estimate the importance of regions such as the Beaufort, East Siberian and Laptev seas in explaining the variance in pan-Arctic summer sea ice area, and over-estimate the percentage of variance explained in regions such as the Barents Sea and parts of the Eurasian basin. Once again, we also see how the regions of highest strength are not necessarily the ones which explain the highest percentage of variance in pan-Arctic sea ice area in the models, which suggests that the models may be over-estimating the spatial extent of the network nodes in

the Eurasian seas; causing them to covary more strongly with other nodes despite having perhaps weaker correlation with the pan-Arctic sea ice area signal.

### 4.3    AO to sea ice teleconnection

We now turn to an investigation of the winter AO to summer sea ice teleconnection. We begin by illustrating how we can use the network framework to exploit this relationship in the observational and reanalysis data by effectively considering both winter

sea-level pressure and summer sea ice concentration networks as individual layers within a *multi-layer* network (Boccaletti et al., 2014). We also briefly investigate whether this teleconnection may be changing over time, before ultimately performing the same analysis for each of the CMIP6 models. A discussion of the results in this section is then presented in Sect. 5.

#### 4.3.1    Observations

In this section we use the temporal component of variability associated with the 'AO node' of the ERA5 sea-level pressure

network to define the time series corresponding to the winter AO (i.e., the dashed time series in Fig. 2). We then generate links between the winter AO and summer sea ice as the temporal covariance (Eq. 2) between this AO time series, and each of the nodes of the summer sea ice concentration networks from each of the observational data sets. In Fig. 10 we use the same concept as the strength maps shown previously, but instead weight each grid cell by the link weight (temporal covariance) between the AO and sea ice concentration node time series. In the first row of Fig. 10 we compute the link weights using the

entire observational period (1979–2020), where we notice a very strong anti-correlation between the winter AO and summer sea ice in the East Siberian Sea across all observational products (standardising the link weight for this node produces correlation coefficients of −0.65, −0.57, and −0.66 for the NASA Team, Bootstrap and OSI-SAF data sets respectively). Furthermore, all of the summer sea ice nodes in the Eurasian–Pacific sector of the Arctic exhibit varying degrees of anti-correlation with the winter AO, while the Atlantic sector shows largely positive covariance, and is particularly strong in the Fram Strait region. If

we then analyse the covariance between the first half (1979–1999) and second half (2000–2020) of the observational record





(second row and third row of Fig. 10 respectively) we notice some interesting patterns. In particular, the correlation across the whole Eurasian–Pacific sector of the Arctic has been more strongly negative since 2000, especially within the Canada basin. The reverse of sign in the Canada basin may not be significant given the moderate degree of positive correlation between 1979–1999, however the strong negative correlation between 2000–2020 implies that positive AO winters (anomalously low

sea-level pressure) now typically lead to anomalously low summer sea ice concentration anomalies across both the eastern and western Arctic, whereas previously this typically only occurred in the eastern Arctic (see also Mallett et al. 2021 for a recent investigation of this changing relationship). It is also worth noting that summer sea ice in the Atlantic sector has generally remained positively correlated with the winter AO over both halves of the observational period – see Sect. 5 for further discussion.

### 4.3.2 CMIP6 models

For each CMIP6 model ensemble we extract the temporal component from the node with the highest strength of each winter sea-level pressure network, and compute the covariance-based link weight with each node of its respective summer sea ice concentration network. In Fig. 11, we show an adaptation of the network comparison metrics shown in Fig. 7. In this case, rather than computing the distance metric $\mathcal{D}$ as the normalised sum of the difference between observational and CMIP6 model

strength maps, we instead create the corresponding 'link maps' for each model ensemble (i.e., the equivalent of the maps shown in Fig. 10), and compute $\mathcal{D}$ relative to the observational link maps. The ARI metric is computed as before, hence ARI values presented in Figs. 7 and 11 are identical. The values reported in Fig. 11 are for link weights computed over the entire period (1979–2020), and with an average distance values of 0.26, 0.27 and 0.26 relative to NASA Team, Bootstrap and OSI-SAF respectively, we can see that the models perform quite poorly at replicating the observed network links between the winter AO

and summer sea ice – recall that for $\mathcal{D} = 0$, the two maps are as dissimilar as a random assignment of link weights to grid cells. The equivalent plots for the periods 1979–1999 and 2000–2020 are shown in supplementary Figs. S1 and S2 respectively. Fig. 12 shows two examples of CMIP6 ensemble member link maps for the winter AO to summer sea ice teleconnection between 1979–2020. The MIROC6 model (r1i1p1f1) was shown in Fig. 8 to be a network which produced relatively similar patterns of summer sea ice variability compared to the observations, and here is one of the models with the highest similarity score in

terms of its AO to sea ice teleconnection, with $\mathcal{D}$ values of 0.43 (NASA Team), 0.43 (Bootstrap), and 0.41 (OSI-SAF). We can see that it also captures the strong negative covariance linkage with the East Siberian Sea, however it over-estimates the connection within the Laptev Sea, and does not capture the negative link with the Beaufort Sea, or strong positive link with the Fram Strait. The EC-Earth3-Veg model (r4i1p1f1) produces the lowest $\mathcal{D}$ scores, at 0.06 (NASA Team), 0.07 (Bootstrap), and 0.06 (OSI-SAF). This is both due to the difference in sign of many of the AO to sea ice node link weights compared to the

observations (e.g., Kara and Beaufort seas), and also due to its inability to represent the similar regions of sea ice variability as the observations. In Fig. 13 we show the average teleconnection link weights between the winter AO and summer sea ice concentration node time series, for both the observations and the average of all CMIP6 ensemble members. Generally, the models agree on the sign of the network links between the winter AO and summer sea ice in the East Siberian and Laptev seas, however the magnitude of this connection is under-estimated on average. The models also do not capture the positive





345 connection with the Kara and Barents seas, and also do not show the same transition to an overall negative connection in the Eurasian–Pacific sectors between 2000–2020. Instead, the Canada basin region remains moderately positively correlated over the entire record.

## 5 Discussion and Conclusions

In this study we used a combination of cluster analysis and complex networks to derive spatio-temporal patterns of variability
350 in northern-hemisphere winter sea-level pressure, and Arctic summer sea ice concentration over the period 1979–2020, and to subsequently understand the spatio-temporal network connectivity between the winter Arctic Oscillation (AO) and summer sea ice cover over the same period. We analysed these patterns in both satellite observational data sets and ERA5 atmospheric reanalysis, and also from 31 of the latest generation of General Circulation Models (GCMs) participating in the most recent phase of the Coupled Model Intercomparison Project (CMIP6). We also introduced two global metrics for comparing patterns
355 of variability between two networks: the Adjusted Rand Index (Hubert and Arabie, 1985) and a network distance metric (Fountalis et al., 2015). Together these allowed us to assess how CMIP6 models perform at replicating the patterns of both winter sea-level pressure, and summer sea ice concentration variability, relative to ERA5 and the observations respectively.

Previous studies (Rigor et al. 2002; Williams et al. 2016) have suggested a mechanism for the winter AO to summer sea ice teleconnection as follows: a positive winter AO (anomalously low mean sea-level pressure) is coincident with (a) a weakening
360 of the Beaufort Gyre, which reduces the amount of west-to-east ice advection, (b) a strengthening of the Transpolar Drift Stream (TDS), which increases ice export out of the Fram Strait, and (c) an increase in cyclonic ice motion in the Eurasian–Pacific sectors of the Arctic, which causes increased ice divergence and facilitates new winter ice formation. Once the melt season begins, these expanses of relatively thin ice are then more susceptible to melting, thus generally leading to anomalously low sea ice area by the end of summer. We have seen in Fig. 10 that the observations support various aspects of this hypoth-
365 esis, by the fact that the strong negative covariance in the East Siberian Sea means that following a positive winter AO, this region typically sees anomalously low sea ice area in the summer, as the ice has undergone thinning and subsequent melting in the spring–summer. The positive covariance in the Fram strait region suggests that following positive AO winters we see an increase in sea ice in this area, which is due to positive AO events strengthening the TDS, resulting in large quantities of ice being advected towards the Atlantic sector (Rigor et al. 2002; Ricker et al. 2018). The fact that we see an overall shift towards
370 more strongly negative covariance between 1979–1999 and 2000–2020 across the whole Eurasian–Pacific sector is likely due to the significant reductions in the thicker multi-year ice cover that have occurred in this region over recent decades (Maslanik et al. 2007; Kwok 2018). Between 1979–1999 the substantially thicker ice cover in the western Arctic was able to withstand the thinning caused by increased ice divergence from a weakened Beaufort Gyre during positive AO events, thus allowing it to survive through the melt season. More recently however, a thinner ice cover means areas of open water are more likely to form
375 during the periods of increased ice divergence in the western Arctic, leading to the growth of new ice which is more susceptible to dramatic ice melt throughout the spring–summer.

The inability of certain CMIP6 models to accurately reflect the dominant regions of winter sea-level pressure or summer sea ice



concentration variability, and their connectivity structure, could be due to a number of factors. In Fig. 6 we saw that, compared to ERA5, on average CMIP6 models replicate the spatial patterns of winter sea-level pressure relatively well, however they
generally over-estimate the magnitude of connectivity over the north Pacific, and under-estimate the magnitude over the Arctic Ocean and the north Atlantic; consistent with previous analysis of CMIP5 models (Jin-Qing et al. 2013; Gong et al. 2016). A recent study by Gong et al. (2019) suggested that the strong north Pacific pattern of variability in GCMs is likely due to the over-estimation of the strength of the stratospheric polar vortex (a persistent feature of models with lower vertical resolutions in their atmospheric components), which causes enhanced coupling of atmospheric circulation between the north Pacific and
north Atlantic.

In terms of summer sea ice, we have seen in Fig. 9 that CMIP6 models show discrepancies in the regions which govern summer sea ice variability, and that they generally under-estimate the contributions from regions such as the Beaufort, East Siberian and Laptev seas (Pacific sector) in explaining pan-Arctic summer sea ice area variability; and similarly over-estimate contributions from the Barents Sea and Eurasian basin (Atlantic sector). The biases in the Atlantic sector are likely related to the model's
over-estimation of the sea ice extent in these regions, as in reality these regions are now largely ice-free in summer (hence the observations show little variability). Meanwhile, biases in the Pacific sector are more likely due to the poor representation of the spatial sea ice thickness distribution in models, which was previously shown to be an issue in CMIP5 models (Stroeve et al., 2014), and also recently for a subset of CMIP6 models (Watts et al., 2021). The sea ice thickness distribution strongly determines how susceptible regions are to melting in the summer (Massonnet et al., 2018), as thicker ice effectively dampens
the amount of energy transfer between the atmosphere and ocean. In Fig. 14 we show the average regional summer sea ice thickness in the Beaufort, East Siberian and Laptev seas from both PIOMAS and 25 of the CMIP6 models used in this study (thickness outputs were not available for BCC-CSM2-MR, CAMS-CSM1-0, CAS-ESM2-0, FGOALS-g3, FIO-ESM-2-0 and CanESM5-CanOE at the time of this study). On average, CMIP6 models report higher average thickness than PIOMAS in each region, which could explain their relatively low contributions to pan-Arctic summer sea ice area variability.
The regional biases in sea ice thickness estimates from CMIP6 models could be related to several factors which determine sea ice transport and hence the ice thickness distribution, including biases in surface winds, ice rheology, and ocean heat fluxes (Stroeve et al. 2014; Watts et al. 2021). Given then that positive winter AO events typically act to pre-condition the ice for increased melting (Williams et al., 2016), models which may perhaps reflect the spatio-temporal patterns of winter sea-level pressure variability well, may still mis-represent the effects of the winter AO on summer sea ice because the ice is too thick,
and subsequently, they therefore under-estimate the amount of variability that these sea ice regions explain in terms of pan-Arctic summer sea ice area. To briefly test this hypothesis, Fig. 15 shows the average CMIP6 winter AO to summer sea ice teleconnection (as in Fig. 13), although this time computed only for a subset of 15 model ensembles (see Table S1) which show the lowest total Root Mean Square Error (RMSE) in terms of their mean summer sea ice thickness relative to PIOMAS in the East Siberian, Laptev, and Beaufort seas. Comparing this with Fig. 13, we notice that when only considering the models with
thinner regional sea ice, the magnitude of covariance between the winter AO and summer sea ice in the East Siberian-Laptev seas is increased, and that between 1979–1999 and 2000–2020 there is evidence of the Beaufort Sea becoming more negatively correlated, although still with a lower magnitude than shown in the observations.



Framing these results in the perspective of dynamical sea ice forecasts, the accuracy of seasonal to inter-annual sea ice pre-
dictions in GCMs ultimately hinges upon their ability to reproduce the physical processes that drive sea ice variability, and
subsequently their ability to reflect the geographic regions which are responsible for explaining the overall variability in sum-
mer sea ice area. In recent years we have seen the improvement in seasonal predictions brought by initialising dynamical
models with observations of sea ice thickness (Chevallier and Salas-Mélia 2012; Doblas-Reyes et al. 2013; Day et al. 2014;
Collow et al. 2015; Bushuk et al. 2017; Allard et al. 2018; Blockley and Peterson 2018; Schröder et al. 2019; Ono et al. 2020;
Balan-Sarojini et al. 2020), therefore reducing sea ice thickness biases in GCMs could be a way to potentially improve the
representation of the winter AO to summer sea ice teleconnection in those models. To more accurately determine the specific
role of regional sea ice thickness variability in this teleconnection, and subsequently isolate the point at which this teleconnec-
tion breaks down across CMIP6 models, the methodology here could be advanced to include causal inference principles (e.g.,
Runge et al. 2019) while also including additional winter–spring and spring–summer climate variables which act as mediators
between winter AO events and the eventual response of summer sea ice; ultimately as a way to begin to bridge the gap between
perfect-model and operational dynamical sea ice forecasts.

*Code availability.*   The following repository contains Python code written by WG, which can be used to access and download CMIP6 data
volumes, as well as to perform the complex networks analysis of all data types: https://github.com/William-gregory/CMIP6. In the same
repository are Python codes to produce ARI and distance metrics for the generated networks.

*Author contributions.*   WG developed the networks code and subsequently ran the analysis for all the observational, reanalysis and model
data. JC originally suggested the idea of applying the complex networks framework to analysing sea ice teleconnections in CMIP6 models and
also provided technical support and input for this paper. MT provided technical support and input for this paper. Both co-authors contributed
to all sections of the paper.

*Competing interests.*   At least one of the (co-)authors is a member of the editorial board of The Cryosphere.

*Acknowledgements.*   WG acknowledges support from the UK Natural Environment Research Council (NERC) (Grant NE/L002485/1). MT
acknowledges support from the NERC "PRE-MELT" (Grant NE/T000546/1) and "EXPRO+ Snow" (ESA AO/1-10061/19/I-EF) projects.



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



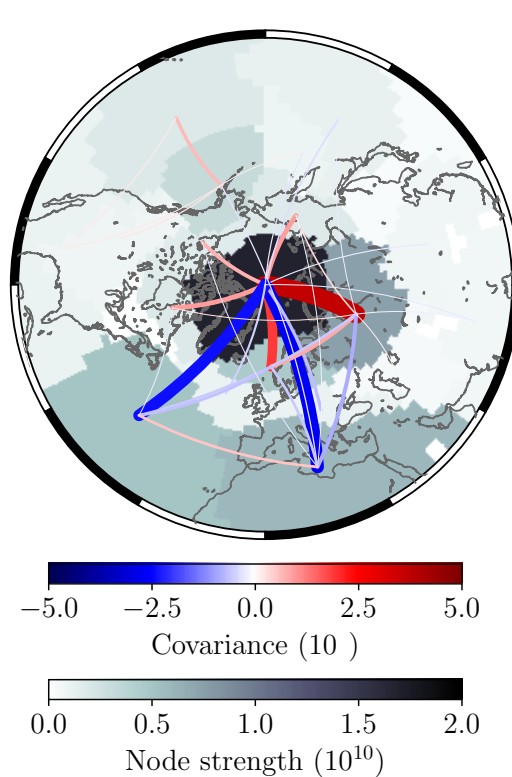

**Figure 1.** Complex network of DJFM mean sea-level pressure from ERA5, computed between 1979–2020. The covariance based link weights are computed from Eq. 2, where the thickness of each link is proportional to the covariance. Only links which have a corresponding p-value < 0.10 are shown here to aid visualisation. The strength of each node is then computed from Eq. 3.





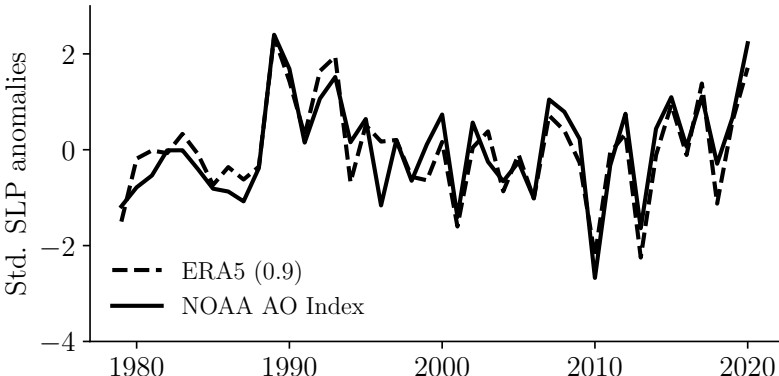

**Figure 2.** The standardised (Std.) temporal component of DJFM mean sea-level pressure (SLP) variability from ERA5 (dashed curve). Temporal components are computed via Eq. 1, where this time series is extracted from the ERA5 network node with the highest strength. The number in parentheses corresponds to the correlation coefficient with the DJFM AO index from the National Oceanic and Atmospheric Administration (NOAA; available from https://www.cpc.ncep.noaa.gov/products/precip/CWlink/daily_ao_index/ao.shtml).

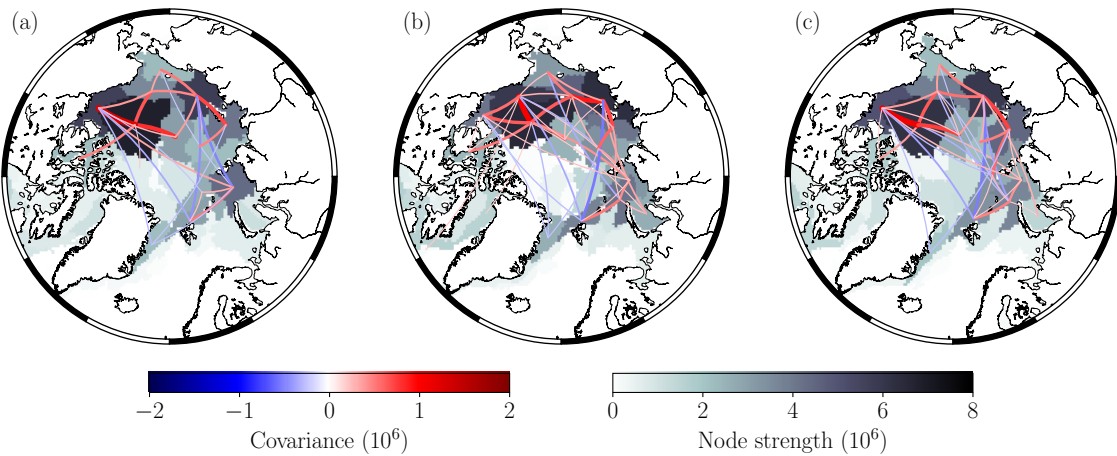

**Figure 3.** Complex networks of JJAS sea ice concentration for (a) NASA Team, (b) Bootstrap, and (c) OSI-SAF data sets, computed between 1979–2020. Only links which have a p-value < 0.10 are shown here to aid visualisation.



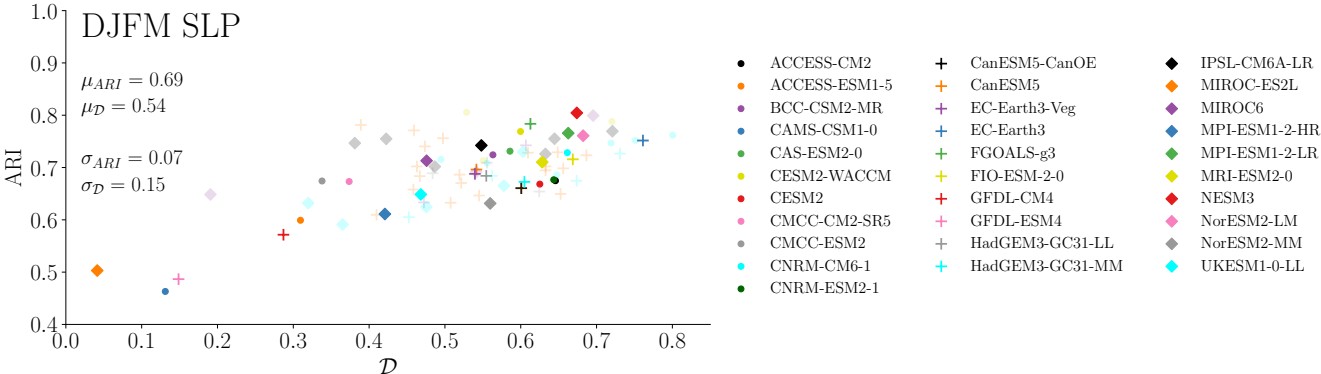

**Figure 4.** ARI and $\mathcal{D}$ metrics for winter sea-level pressure (SLP) networks computed between 1979–2020 for every ensemble member for 31 different CMIP6 models (74 realisations), relative to ERA5 atmospheric reanalysis. The semi-transparent colours represent individual ensemble members (where the number of ensembles is greater than 1), and the opaque colours are the mean of all ensemble members. The mean and standard deviation across all points are given by $\mu$ and $\sigma$ respectively.

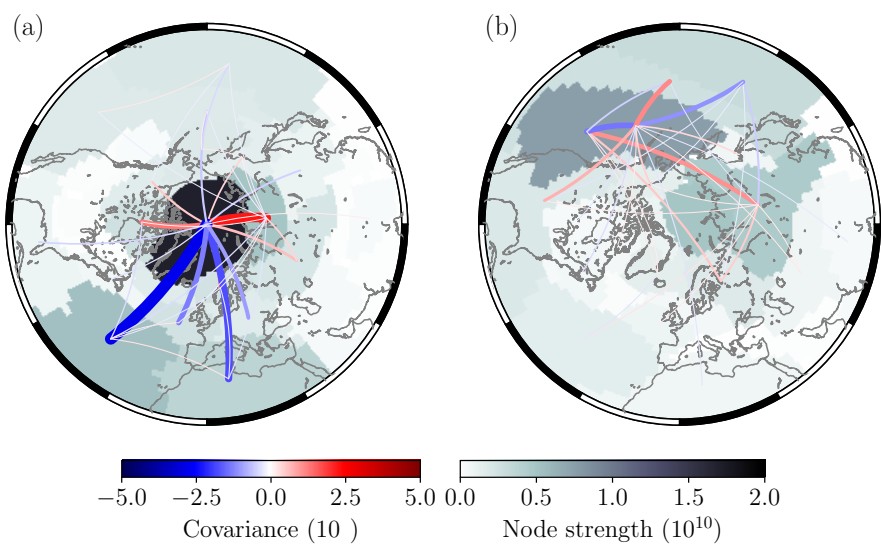

**Figure 5.** Winter sea-level pressure networks from (a) CNRM-CM6-1 (ensemble: r6i1p1f2) and (b) MIROC-ES2L (ensemble: r1i1p1f2), computed between 1979–2020. The CNRM-CM6-1 model produces ARI and $\mathcal{D}$ values 0.76 and 0.80 respectively, while the MIROC-ES2L model produces values 0.50 and 0.04 respectively. Only links which have a corresponding p-value $< 0.10$ are shown here to aid visualisation.

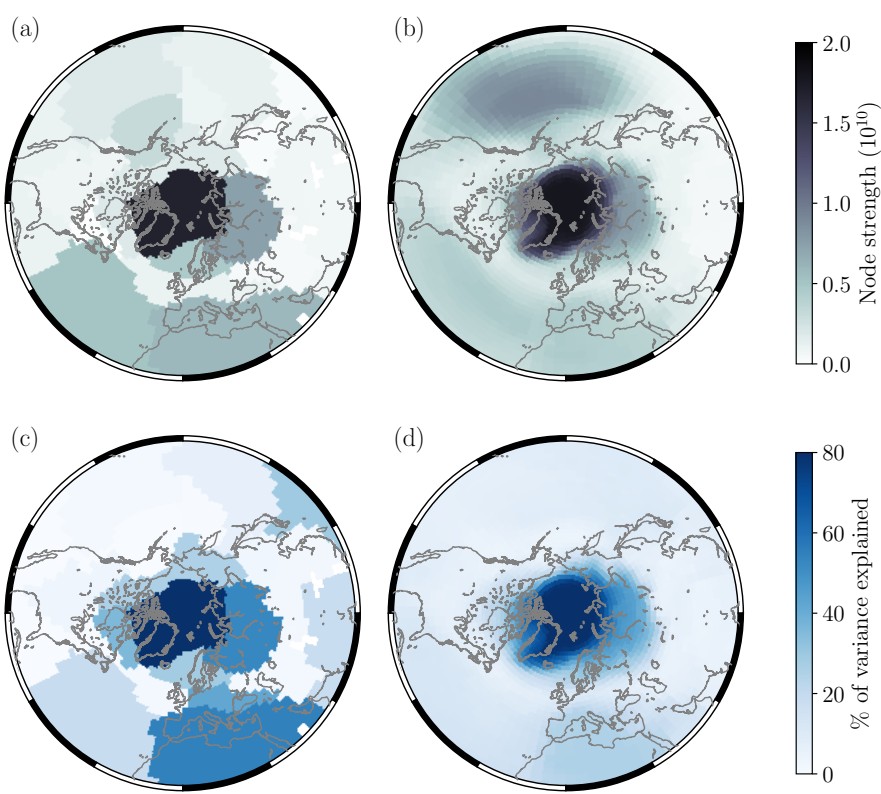

**Figure 6.** (a)-(b) The spatial patterns of winter sea-level pressure variability (node strengths) from (a) ERA5, and (b) the average of all 74 CMIP6 model ensemble members. (c)-(d) The percentage of variance in mean northern-hemisphere winter sea-level pressure explained (e.g., Björnsson and Venegas 1997) by network nodes in (c) ERA5, and (d) the average of all 74 CMIP6 model ensemble members.





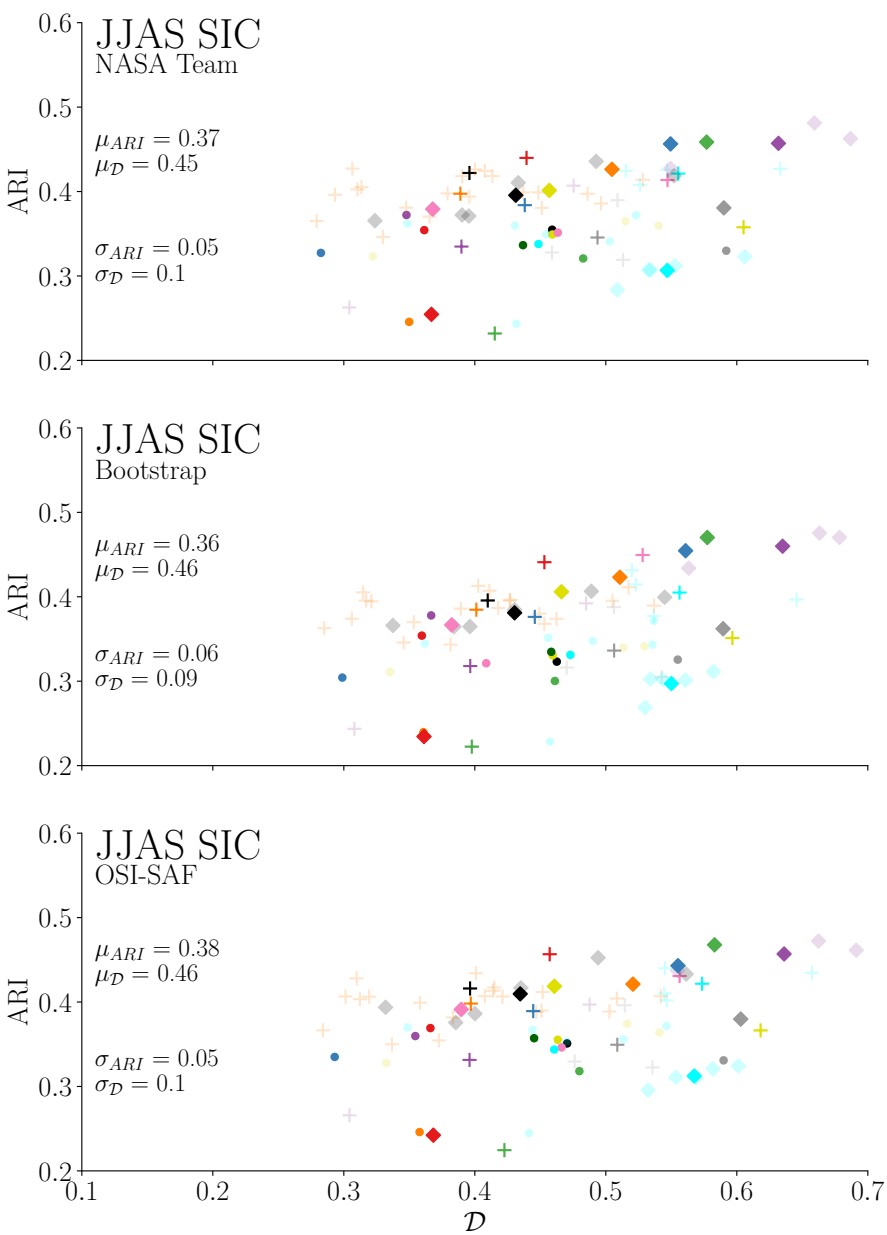

**Figure 7.** ARI and $\mathcal{D}$ metrics for summer sea ice concentration (SIC) networks computed between 1979–2020 for every ensemble member for 31 different CMIP6 models (74 realisations). ARI and $\mathcal{D}$ are computed relative to NASA Team (top), Bootstrap (middle) and OSI-SAF (bottom) observational networks. The symbols and colours of each point are consistent with Fig. 4.

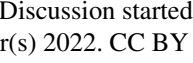



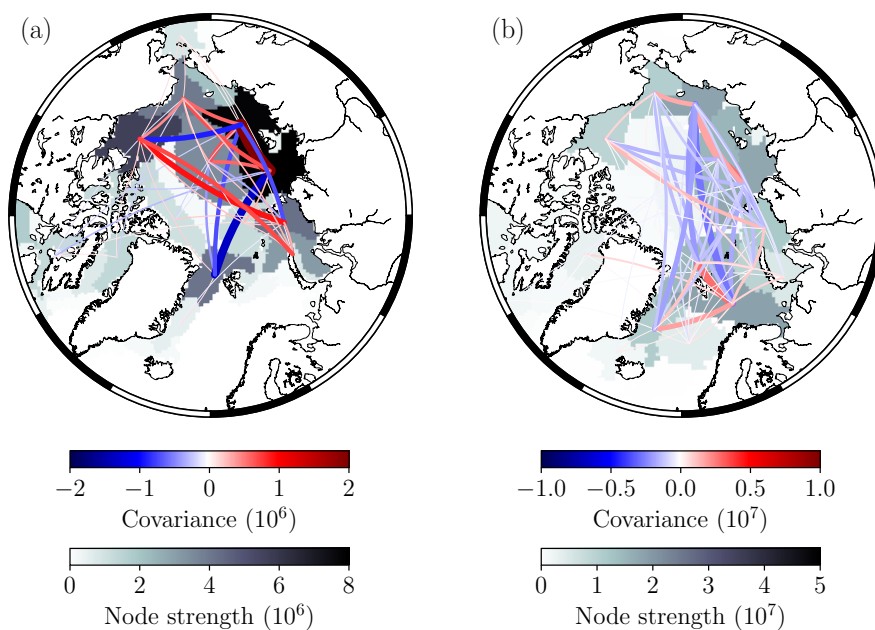

**Figure 8.** Summer sea ice concentration networks from (a) MIROC6 (ensemble: r1i1p1f1) and (b) CAMS-CSM1-0 (ensemble: r1i1p1f1), computed between 1979–2020. The MIROC6 model produces average ARI and $\mathcal{D}$ values of 0.48 and 0.66 respectively (average of metrics computed relative to NASA Team, Bootstrap and OSI-SAF networks), while the CAMS-CSM1-0 model produces average values of 0.32 and 0.29 respectively. Only links which have a corresponding p-value $< 0.10$ are shown here to aid visualisation.

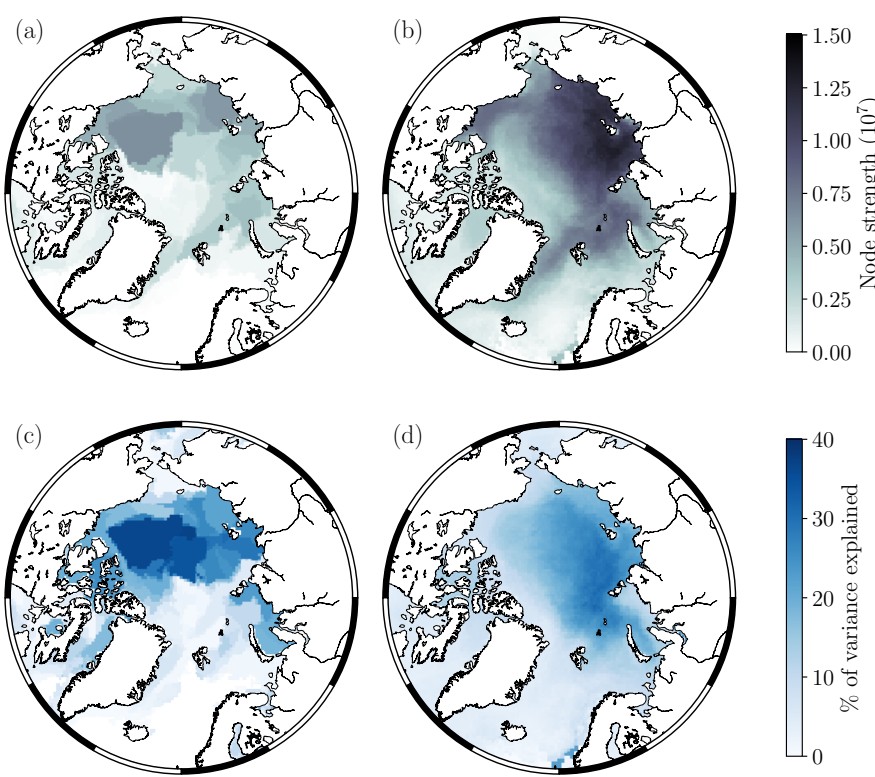

**Figure 9.** (a)-(b) The average spatial patterns of summer sea ice concentration variability (node strengths) from (a) the three observational data sets, and (b) all 74 CMIP6 model ensemble members. (c)-(d) The percentage of variance in pan-Arctic summer sea ice area explained by network nodes in (c) the three observational data sets, and (d) the average of all 74 CMIP6 model ensemble members.

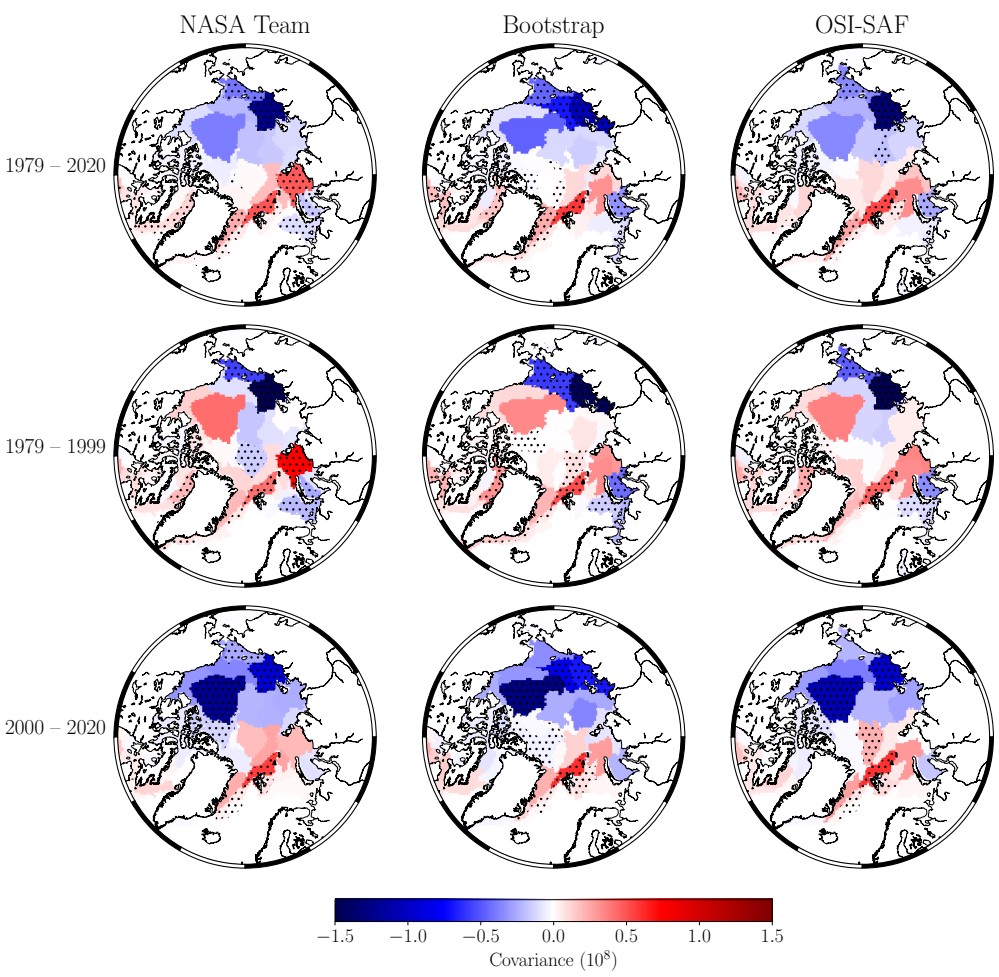

**Figure 10.** Network link weight between the DJFM ERA5 'AO node' (dashed time series from Fig. 2), and each of the JJAS sea ice concentration network nodes, computed between (first row) 1979–2020, (second row) 1979–1999, and (third row) 2000–2020. The columns from left to right show the corresponding maps for NASA Team, Bootstrap, and OSI-SAF data sets. Stippling denotes links with p-values $< 0.05$.



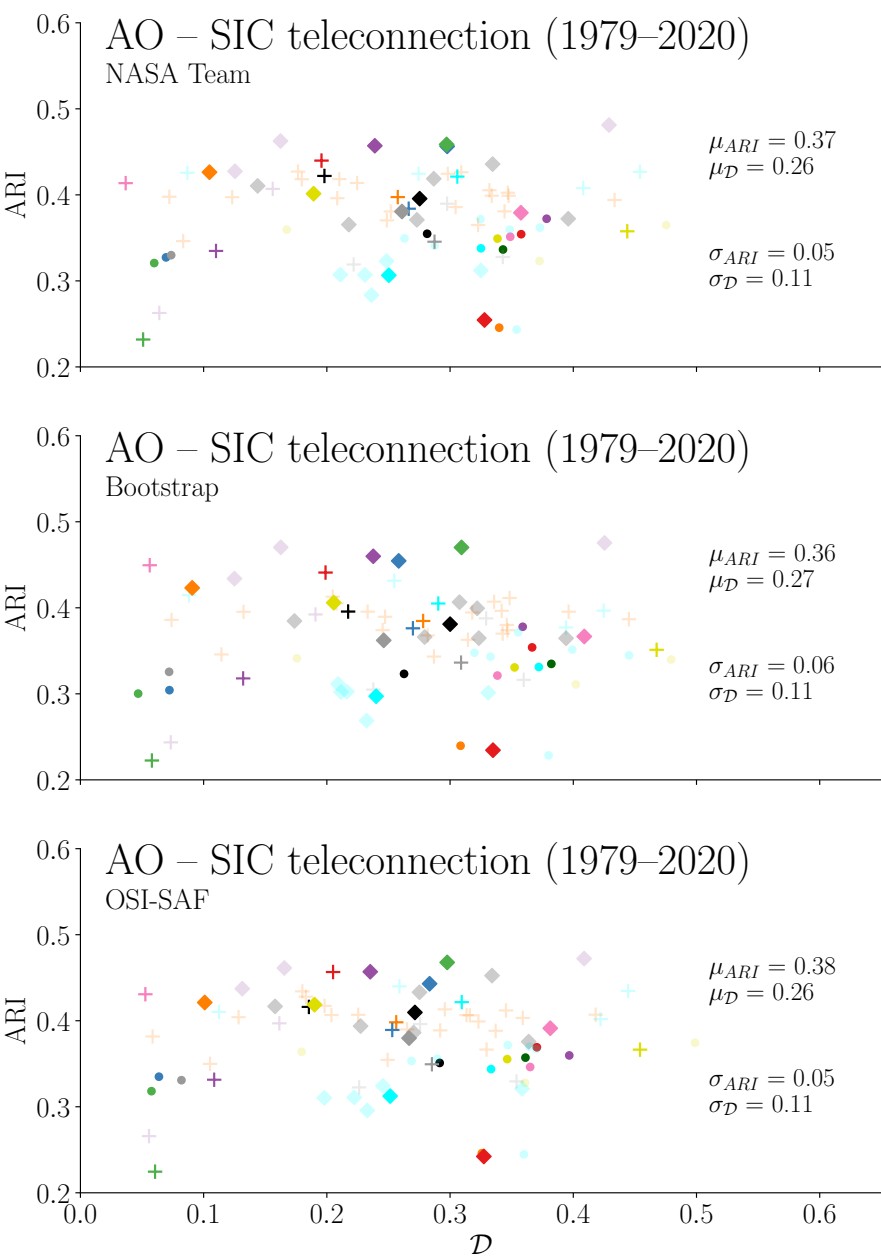

**Figure 11.** ARI and $\mathcal{D}$ metrics for comparing observation and CMIP6 model summer sea ice concentration networks and the winter AO to summer sea ice teleconnection, for every ensemble member for 31 different CMIP6 models (74 realisations). ARI and $\mathcal{D}$ are computed relative to NASA Team (top), Bootstrap (middle), and OSI-SAF (bottom) observational networks. Network distance values ($\mathcal{D}$) are computed from observation and model 'link maps' as shown in Fig. 10. The symbols and colours of each point are consistent with Fig. 4.


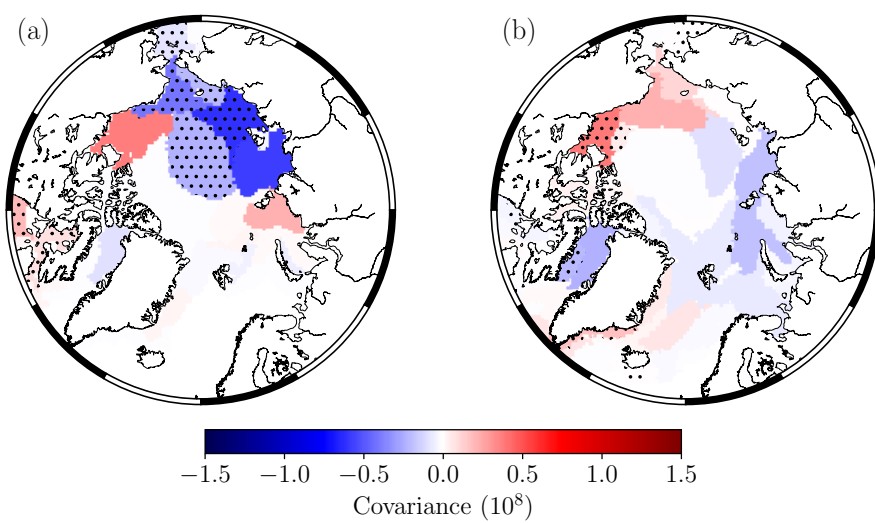

**Figure 12.** Covariance-based link weights between the winter AO node time series and each node of the summer sea ice concentration network, between 1979–2020 for (a) MIROC6 (ensemble: r1i1p1f1) and (b) EC-Earth3-Veg (ensemble: r4i1p1f1). The MIROC6 model produces average ARI and $\mathcal{D}$ values 0.47 and 0.56 respectively, while the EC-Earth3-Veg model produces values 0.26 and 0.97 respectively. Stippling denotes links with p-values $< 0.05$.



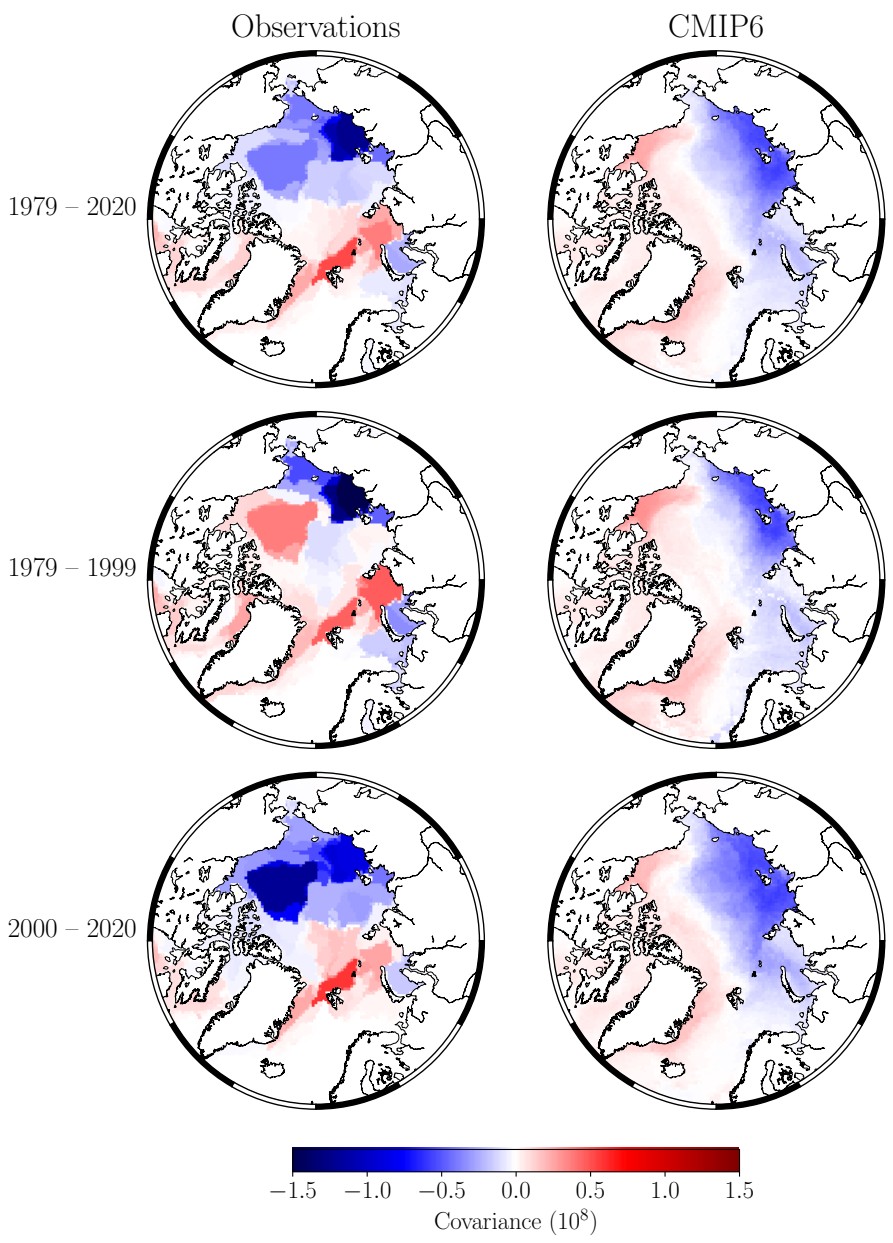

**Figure 13.** The average covariance-based link weights between the winter AO node time series and each node of the summer sea ice concentration networks, across both the observations (left column) and all CMIP6 model ensemble members (right column). Each row shows the link weights computed over different parts of the time series record: 1979–2020 (first row), 1979–1999 (second row) and 2000–2020 (third row).

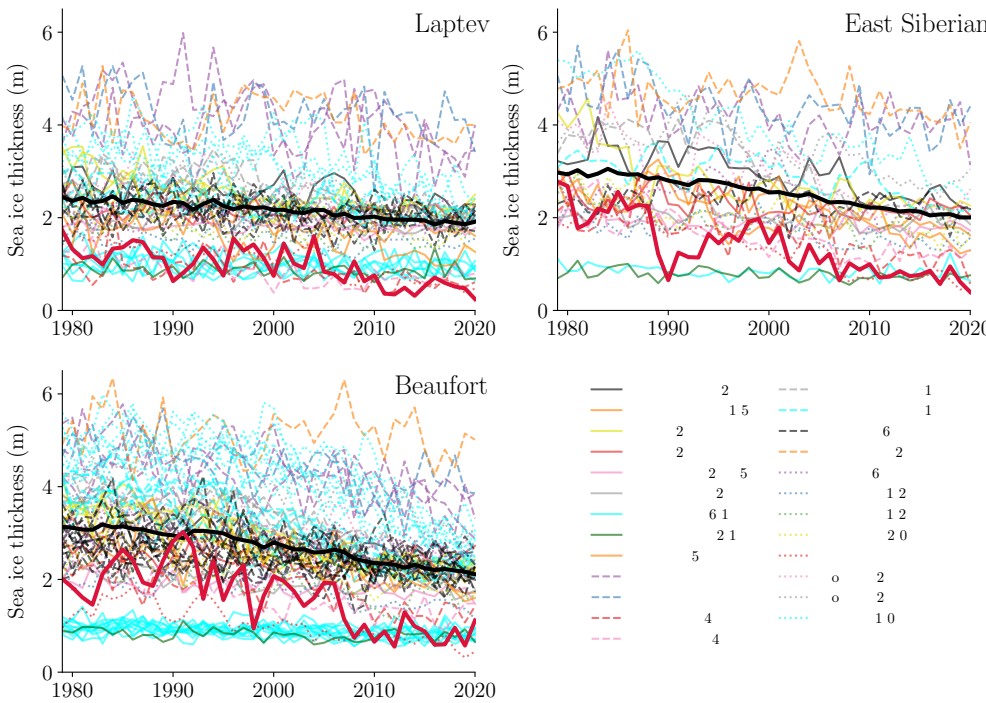

**Figure 14.** The average summer sea ice thickness from 25 CMIP6 models (49 realisations) and PIOMAS, in the Laptev Sea (top left), East Siberian Sea (top right) and Beaufort Sea (bottom). PIOMAS is given by the bold red curve, and the average of all CMIP6 models is given by the bold black curve.

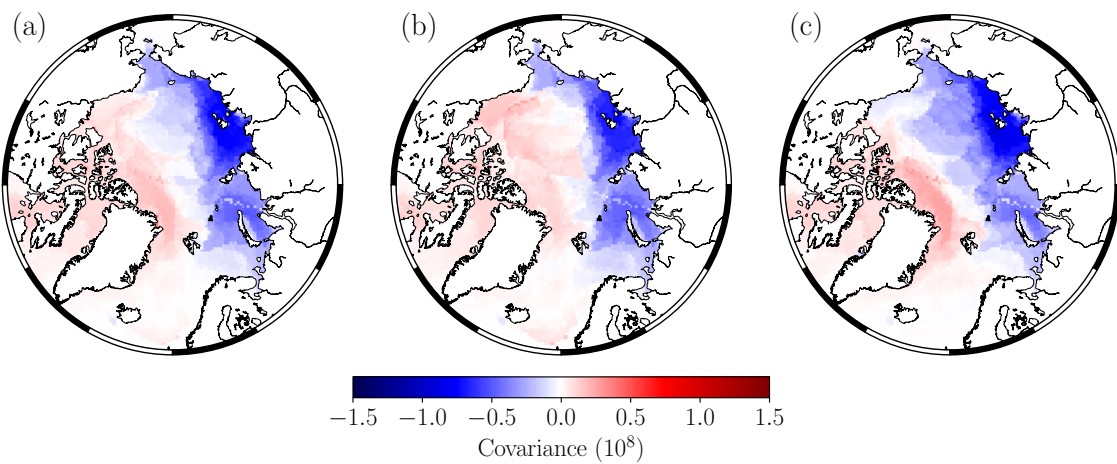

**Figure 15.** Average winter AO to summer sea ice teleconnection for 15 CMIP6 model ensembles with the lowest average Root Mean Square Error (RMSE) in mean sea ice thickness relative to PIOMAS in the East Siberian, Laptev, and Beaufort seas. (a) Links computed between 1979–2020, (b) 1979–1999, (c) 2000–2020.



**Table 1.** CMIP6 models used in this study.

| Model | No. Ensembles |
| --- | --- |
| ACCESS-CM2 | 1 |
| ACCESS-ESM1-5 | 1 |
| BCC-CSM2-MR | 1 |
| CAMS-CSM1-0 | 1 |
| CanESM5-CanOE | 1 |
| CanESM5 | 20 |
| CAS-ESM2-0 | 1 |
| CESM2 | 1 |
| CESM2-WACCM | 3 |
| CMCC-CM2-SR5 | 1 |
| CMCC-ESM2 | 1 |
| CNRM-CM6-1 | 6 |
| CNRM-ESM2-1 | 1 |
| EC-Earth3 | 1 |
| EC-Earth3-Veg | 2 |
| FGOALS-g3 | 1 |
| FIO-ESM-2-0 | 1 |
| GFDL-CM4 | 1 |
| GFDL-ESM4 | 1 |
| HadGEM3-GC31-LL | 3 |
| HadGEM3-GC31-MM | 4 |
| IPSL-CM6A-LR | 6 |
| MIROC6 | 3 |
| MIROC-ES2L | 1 |
| MPI-ESM1-2-HR | 1 |
| MPI-ESM1-2-LR | 1 |
| MRI-ESM2-0 | 1 |
| NESM3 | 1 |
| NorESM2-LM | 1 |
| NorESM2-MM | 1 |
| UKESM1-0-LL | 5 |





**Table 2.** Synthetic example of cell clusters to illustrate the concept of the Rand Index (see Sect. 3.2.1), after Rand (1971).

| Cell pairs | ab | ac | ad | ae | af | bc | bd | be | bf | cd | ce | cf | de | df | ef | Total |
|---|---|---|---|---|---|---|---|---|---|---|---|---|---|---|---|---|
| Together in both | ✓ | | | | | | | | | | | | | | | 1 |
| Separate in both | | | ✓ | ✓ | ✓ | | ✓ | ✓ | ✓ | | ✓ | ✓ | | ✓ | | 9 |
| Mixed | | ✓ | | | | ✓ | | | | ✓ | | | ✓ | | ✓ | 5 |