# Peer review of "Network connectivity between the winter Arctic Oscillation and summer sea ice in CMIP6 models and observations"

_The Cryosphere, 2021_

## Referee Comment (RC1)

**Review of „Network connectivity between the winter Arctic Oscillation and summer sea ice in CMIP6 models and observations" by Gregory et al.**

**General comments**
This study by Gregory et al. investigates the spatio-temporal covariability of the winter Arctic Oscillation (AO) and summer Arctic sea ice and its representation in CMIP6 models with a complex network approach. This is interesting for the predictability of summer sea ice in the light of a suggested physical mechanism between the winter AO to summer sea ice, supported by the observations. The authors point out discrepancies between the covariability of winter AO and summer sea ice in observations/reanalyses and CMIP6 models, particularly also in the transition from earlier to more recent years, and hypothesize that these could be due to regional biases in sea ice thickness.
This is a carefully performed and well-presented study and a valuable addition to previous literature on mechanisms for sea ice predictability. The paper is clearly structured and well written. The methods are well explained, so that also someone who is not familiar with complex networks can easily follow and understand the results. Previous literature is thoroughly cited. Overall, I recommend the paper for publication after consideration of the comments outlined below.

**Specific comments**
- As you write in L66 ff., climate network analysis is an useful addition to more conventional statistical approaches. I find the climate network analysis very intriguing, but (being new to the method) it is not entirely clear to me why you chose it here and what is the added value to more conventional statistics. Would you expect to arrive at the same conclusions with e.g. an EOF analysis?

- To L124 ff.: Is the number of network nodes N determined by the clustering algorithm or is it prescribed? What would be a typical number for N?

- Equation 2: I understand that the grid cells need to be weighted by their grid cell area. However, I find it a bit puzzling that for the polar grid the weights should be the square-root of the cell area, while for a regular grid the weights are a non-dimensional number between 0 and 1. Would it be possible to introduce normalized area-weights, such that there is no [km] in the unit and the anomaly time series (and with that also the covariance and node strengths) are not dependent on the node size? This could then maybe also resolve the issue that a high node strength does not necessarily coincides with large explained variance, as you explain in L252 ff.?

- Relating to what you write in L175-L187 (which is a very helpful paragraph), I am still wondering how much agreement/disagreement between model simulations (even if they had perfect model physics) and observations you can expect, simply because of internal variability and the time series of 42 years being relatively short, even more so when you look at the first and second half of the time series separately. What could be helpful in this regard is a comparison of the observations to a single-model initial-condition large ensemble, which can give you a good indication of the range of internal variability within the model. The CanESM5 model with its 20 members comes closest to a large ensemble. Looking at the ARI and D values of the CanESM5 ensemble members gives an idea of the expected spread just due to internal variability. I don't have a concrete suggestion for changing the manuscript, but I think this point should be kept in mind when describing and discussing the results.

- In all Figures showing covariance and node strengths, please add the units. Moreover, in the Figures for SLP (Figs. 1 and 5), I don't see how the orders of magnitude for covariance (10) and node strength ($10^{10}$) fit together. As the node strength is the sum of the absolute value of all its associated link weights, it shouldn't be orders of magnitude higher than the individual link weights/covariance, I would assume?

- Figure 8: The two sub-figures are a bit difficult to compare with the different axes ranges/color scales. If you keep the different ranges (which I can see is also useful for comparing Fig. 8a) with the observations), I still think that the range for Fig. 8b) could be

capped at lower values, as right now the colors are very light (which is misleading at first glance).

- In the caption of Figure 12, the values of ARI and D for the individual model simulations don't seem plausible/don't fit to what you write in the main text.
- Figure 14: I don't understand the meaning of the numbers in the legend(?) in the lower right panel of the figure. A proper legend with the names of individual models, the model mean and PIOMAS would be helpful.

**Technical corrections**
- L1: Suggest changing "proceeding" to "following", "subsequent" or similar
- L9: Suggest adding comma before "respectively" (here and at other occasions in the manuscript)
- L14: "the north Africa" → "the north of Africa", "north Africa", or "northern Africa"
- L56: When reading, I was wondering why only 31 of the CMIP6 models, which you explain later in the Data section. Very optional, but you could consider omitting the number of 31 in the introduction to avoid questions at this place.
- L109: Remove comma: "sea ice concentration and mean sea level pressure"
- L117, Table 1 and other occasions: "ensembles" → "ensemble members". In my understanding, a single simulation is denoted as an ensemble member, while the individual simulations together form a (single-model or multi-model) ensemble. If you change this, please be sure to stay consistent throughout the manuscript.
- Eq. 2: Suggest writing down that E[...] is the expected/mean value.
- L245: Remove comma: "north Africa and southern Europe"
- L298: Suggest changing heading from "Observations" to "Observations/Reanalyses"
- L430: JC → JS

---

## Author Comment (AC1)

**Review of "Network connectivity between the winter Arctic Oscillation and summer sea ice in CMIP6 models and observations" by Gregory et al.**

**General comments**

This study by Gregory et al. investigates the spatio-temporal covariability of the winter Arctic Oscillation (AO) and summer Arctic sea ice and its representation in CMIP6 models with a complex network approach. This is interesting for the predictability of summer sea ice in the light of a suggested physical mechanism between the winter AO to summer sea ice, supported by the observations. The authors point out discrepancies between the covariability of winter AO and summer sea ice in observations/reanalyses and CMIP6 models, particularly also in the transition from earlier to more recent years, and hypothesize that these could be due to regional biases in sea ice thickness.

This is a carefully performed and well-presented study and a valuable addition to previous literature on mechanisms for sea ice predictability. The paper is clearly structured and well written. The methods are well explained, so that also someone who is not familiar with complex networks can easily follow and understand the results. Previous literature is thoroughly cited. Overall, I recommend the paper for publication after consideration of the comments outlined below.

We would like to thank the reviewer for their kind words and for the time they have given to review this work. Their comments have been very insightful and have certainly strengthened this manuscript. Please see our responses below.

**Specific comments**

- As you write in L66 ff., climate network analysis is an useful addition to more conventional statistical approaches. I find the climate network analysis very intriguing, but (being new to the method) it is not entirely clear to me why you chose it here and what is the added value to more conventional statistics. Would you expect to arrive at the same conclusions with e.g. an EOF analysis?

  This is a great question. In some specific test cases we would indeed find that the networks approach and EOF analysis produce very similar results. For example, the dominant spatial patterns of variability seen in the network strength map of one particular model ensemble member are likely to also be present in the first few leading EOF maps of that same ensemble member. The purpose of using the networks method in this paper was two-fold. On the one hand, we wanted to use this paper to bring attention to the method itself, as complex networks have become increasingly used in climate science over the last decade, although perhaps not so much in cryospheric/polar domains. Furthermore, it provides a nice initial framework which can be modified for deeper analysis of climatological interactions, such as causal inference techniques. On the other hand, the networks approach does provide some advantages over EOF analysis. For example, EOF analysis is variance greedy, so each of the (EOF) derived modes reflect the direction along which the data exhibit the largest variance, and so patterns of lower variance are inherently masked. Furthermore, it is also perhaps cumbersome to convey how each of these EOF modes are interrelated, while in the networks framework we do not mask patterns of lower variance, and we can also neatly summarise the interconnected nature between nodes(modes) with the strength and link maps. For a more in-depth response to this discussion point, we have provided additional information in the response to reviewer 2's first comment.

- To L124 ff.: Is the number of network nodes N determined by the clustering algorithm or is it prescribed? What would be a typical number for N?

  The number of network nodes N is determined by the clustering algorithm. This is specific to our chosen algorithm, which is a grid-based clustering approach whereby each node tries to continuously grow in spatial extent, so long as the correlations between all points within the node are above a specific threshold. Once the node cannot grow anymore, the next available point is selected, and the process repeated (available point meaning a grid cell that does not already belong to a network node). Alternative clustering algorithms such as K-means require N to be

defined a-priori. The sea-level pressure networks typically see N in the range of 20 to 30, while the number of sea ice concentration network nodes can be typically on the order of 80 to 100.

- Equation 2: I understand that the grid cells need to be weighted by their grid cell area. However, I find it a bit puzzling that for the polar grid the weights should be the square-root of the cell area, while for a regular grid the weights are a non-dimensional number between 0 and 1. Would it be possible to introduce normalized area-weights, such that there is no [km] in the unit and the anomaly time series (and with that also the covariance and node strengths) are not dependent on the node size? This could then maybe also resolve the issue that a high node strength does not necessarily coincides with large explained variance, as you explain in L252 ff.?

  If we understand correctly, you are happy with the weighting of the lat-lon gridded (sea-level pressure) data -- which is a form of normalised area weighting, however your concerns lie primarily with the weighting of the polar stereographic (sea ice concentration) data, and you would like to see a similar kind of normalised area weighting to these data?

  We believe that the current approach is justified because in either the lat-lon, or polar stereo case, the weighting is simply a means to ensure that data points contribute proportionally to the eventual network node anomaly time series, and hence temporal covariance between network nodes. Normalising the area weights for the polar stereo case would produce the same network node anomaly time series as the current approach, but there would just be a difference of scaling. To remove the relationship between covariance and node size we could simply standardise each network node anomaly time series (at which point the links between nodes become the correlation coefficient, as opposed to covariance). It is worth pointing out that both the sea-level pressure and sea ice concentration data show examples of where high node strength does not necessarily coincide with large explained variance and we feel that this result in itself is interesting as it suggests that CMIP6 models produce sea ice concentration and/or sea-level pressure fields which are more homogeneous than the observations (recall that an individual network node represents a geographic region which has behaved homogeneously over the length of the time series record, so the larger the node, the larger the region of homogeneity).

- Relating to what you write in L175-L187 (which is a very helpful paragraph), I am still wondering how much agreement/disagreement between model simulations (even if they had perfect model physics) and observations you can expect, simply because of internal variability and the time series of 42 years being relatively short, even more so when you look at the first and second half of the time series separately. What could be helpful in this regard is a comparison of the observations to a single-model initial-condition large ensemble, which can give you a good indication of the range of internal variability within the model. The CanESM5 model with its 20 members comes closest to a large ensemble. Looking at the ARI and D values of the CanESM5 ensemble members gives an idea of the expected spread just due to internal variability. I don't have a concrete suggestion for changing the manuscript, but I think this point should be kept in mind when describing and discussing the results.

  This is a great point, and certainly an interesting topic that is worth including in the manuscript. In our case the 20 CanESM5 ensemble members actually comprise two groups of 10 members with different physics (group 1: i1p1f1, group2: i1p2f1). While any analysis of these individual groups while likely produce an under-estimate of the spread in ARI and D from internal variability, we agree that a discussion on this should be included at the least. We have added some content and figures to the revised manuscript to highlight this.

- In all Figures showing covariance and node strengths, please add the units. Moreover, in the Figures for SLP (Figs. 1 and 5), I don't see how the orders of magnitude for covariance (10) and node strength ($10^{10}$) fit together. As the node strength is the sum of the absolute value of all its associated link weights, it shouldn't be orders of magnitude higher than the individual link weights/covariance, I would assume?

  We have double checked this, and it seems ok. In Fig. 1 for example, the magnitude of the first 5 strongest links belonging to the AO node are as follows: 3.7e9, 2.6e9, 2.4e9, 1.9e9, 0.98e9. Hence the sum of all of these links is only slightly over 1.16e10. We have also added units to each of the plots.

- Figure 8: The two sub-figures are a bit difficult to compare with the different axes ranges/color scales. If you keep the different ranges (which I can see is also useful for comparing Fig. 8a) with the observations), I still think that the range for Fig. 8b) could be capped at lower values, as right now the colors are very light (which is misleading at first glance).
  Thank you for the suggestion, we have edited this figure to cap the range of 8b accordingly.
- In the caption of Figure 12, the values of ARI and D for the individual model simulations don't seem plausible/don't fit to what you write in the main text.
  Thank you for spotting this. The values in the figure caption are incorrect and have now been edited to reflect the main text.
- Figure 14: I don't understand the meaning of the numbers in the legend(?) in the lower right panel of the figure. A proper legend with the names of individual models, the model mean and PIOMAS would be helpful.
  Apologies, this appears to have been a bug in the file upload. The original image contained the individual model names in the legend and no numbers. We have amended this and added the model means to the legend as per your suggestion.

**Technical corrections**
- L1: Suggest changing "proceeding" to "following", "subsequent" or similar
  Thank you, we have opted for "following"
- L9: Suggest adding comma before "respectively" (here and at other occasions in the manuscript)
  Agreed, we have edited this throughout the manuscript.
- L14: "the north Africa" → "the north of Africa", "north Africa", or "northern Africa"
  Thank you, we have edited this to "north Africa"
- L56: When reading, I was wondering why only 31 of the CMIP6 models, which you explain later in the Data section. Very optional, but you could consider omitting the number of 31 in the introduction to avoid questions at this place.
  Thanks for the tip, we have removed the explicit reference to 31 models in the introduction
- L109: Remove comma: "sea ice concentration and mean sea level pressure"
  Thank you, this has been edited
- L117, Table 1 and other occasions: "ensembles" → "ensemble members". In my understanding, a single simulation is denoted as an ensemble member, while the individual simulations together form a (single-model or multi-model) ensemble. If you change this, please be sure to stay consistent throughout the manuscript.
  Thank you, we have changed this throughout the manuscript.
- Eq. 2: Suggest writing down that E[…] is the expected/mean value.
  Agreed, this is more clear.
- L245: Remove comma: "north Africa and southern Europe"
  Thank you
- L298: Suggest changing heading from "Observations" to "Observations/Reanalyses"
  Agreed
- L430: JC → JS
  Thank you!

---

## Author Comment (AC2)

**Review: Network connectivity between the winter Arctic Oscillation and summer sea ice in CMIP6 models and observations**

Gregory et al. 2022

The Cryosphere

**General comments**

This paper introduces complex networks as a relatively new method to the field of climate science, and applies it to highlight differences between observations and models in the Arctic oscillation, regional sea ice variability and the links between the two. This new method has some interesting potential, and makes the paper a valuable contribution to the field even before considering the later results and insights. The general subject of the paper, Arctic oscillation and its link to sea ice, is one of considerable importance, and this paper offers an interesting angle on the mismatch between models and observations.

In general the paper is well written, with well constructed figures and presents the context for its findings in terms of previous research well. Overall I think that only a few minor edits would be needed before it would be ready for publication.

We would like to thank the reviewer for their kind words and for their time given to review this manuscript. Their comments have been thought-provoking and have certainly helped to improve this manuscript. Please see our responses below.

**Specific comments**

While I appreciate the use of complex networks in this paper, I'm still a little unsure of what the advantages and disadvantages are of their use compared to more traditional methods like EOF or maximum covariance analysis, even after skimming through Donges et al. 2015. I think that including at least a sentence or two more on this topic would be beneficial. In an ideal world it would be great to directly compare the results (e.g. those in figure 13) with equivalents from EOF analysis, but I appreciate that this would be a substantial effort and could be outside the scope of this paper. To my knowledge, the results in figure 13 are very similar if the AO index from EOF analysis is regressed on observed SIC and that in the CESM1 large ensemble (https://doi.org/10.1175/JCLI-D-20-0958.1, figure 12a and b - not implying that this should be cited, a similar result is probably presented in more detail in some other paper).

This is a great point, and one which was similarly raised by reviewer 1. We will expand a bit on our answer here: In some specific test cases we would indeed find that the networks approach and EOF analysis produce very similar results. For example, the dominant spatial patterns of variability seen in the network strength map of one particular model ensemble member are likely to also be present in the first few leading EOF maps of that same ensemble member. The purpose of using the networks method in this paper was two-fold. On the one hand, we wanted to use this paper to bring attention to the method itself, as complex networks have become increasingly used in climate science over the last decade, although perhaps not so much in cryospheric/polar domains. Furthermore, it provides a nice initial framework which can be modified for deeper analysis of climatological interactions (in some ways this relates to your final comment below, regarding future work). For example, in this work we have shown somewhat simple network constructions, with some examples of intra-layer connections (i.e., SIC to SIC, or SLP to SLP), and a single inter-layer connection (AO to SIC). Through a bit of graphical manipulation, the same complex networks methodology we use here can be exploited to provide a very visually intuitive understanding of climate interactions with multiple intra- and inter-layer connections across any number of network layers (e.g., Figure 6 in Donges et al., 2015); a feature which would perhaps be quite difficult to achieve with EOF analysis. What's more, the definition of the link weights is very flexible, for example we could in theory apply the principles of causal inference to our networks to prune certain connections, while also accounting for additional factors such as non-linearity. While non-linear forms of EOF analysis do exist, establishing spatio-temporal causality from EOF modes, to our knowledge, is not straightforward.

Some more technical advantages of networks over EOF analysis for example, are that a) EOF analysis is variance greedy, so each of the (EOF) derived modes reflect the direction along which the data exhibit the largest variance, and so patterns of lower variance are inherently masked. b) EOF analysis imposes orthogonality constraints, such that each mode of variability must exist in a direction which is orthogonal to all other modes. In contrast, the networks approach has neither of these constraints. Furthermore, it is also perhaps cumbersome to convey how each of these EOF modes are interrelated, while in the networks framework we can also neatly summarise the interconnected nature between nodes(modes) with the strength and link maps (another motivation for why we chose this methodology). Perhaps an advantage of EOF analysis is that the solution is well defined. Our clustering algorithm here is one of many such algorithms and often what defines a cluster can be quite heuristic. Hence while our algorithm produces a deterministic solution for a network, there are likely many other viable definitions depending on the clustering algorithm chosen. Our algorithm was chosen based on successful previous analysis of ocean teleconnections by Fountalis et al., 2014 (https://doi.org/10.1007/s00382-013-1729-5).

With regards to your final comment above (also thank you for bringing the recent paper by Clancy et al to our attention!), it is perhaps worth mentioning that, in general, the goal of complex networks analysis is to provide a framework in which to construct simple illustrations of complex systems, such as the Earth's climate. While Figure 13 does provide a result which would appear similar to regressing the AO index on gridded SIC, it is in some ways more simple than that. The clustering component of the methodology acts to emphasise the key regions of variability and their spatial extent by dimensionality reduction, so that we can ignore small-scale heterogeneities which might otherwise be related to small-scale processes and/or noise. Indeed, while this is one part of the result, we believe the strength in the methodology also comes from our ability to quantify similarities in spatio-temporal structures by the ARI and D metrics.

In summary, we agree that it is worthwhile including some discussion in the introduction about the similarities/differences between EOF analysis and complex networks, and highlight our reasons for choosing the networks approach. We have incorporated this into the revised manuscript.

Another interesting comparison with EOF based mechanisms is to look at the AO sea level pressure pattern in observations (https://www.cpc.ncep.noaa.gov/products/precip/CWlink/daily_ao_index/ao.loading.shtml). Is there a clear explanation for why the AO doesn't have a strong negative link to the north Pacific in figure 1, while the AO from EOF analysis is so strongly connected to that region?

The discrepancies are possibly due to different handling of the input data here. The image in the link above appears to be the leading EOF of monthly-mean geopotential height anomalies for the period 1979 – 2000 (for all months of the year), while our analysis was only across winter months, and between 1979 – 2020. In the image below we compute the leading EOF using the same ERA5 data as we use in the manuscript (winter SLP anomalies between 1979 – 2020), where we can see the comparably weaker variability in the Pacific sector compared to the Atlantic.

[Figure]

Line 128 - Could the authors clarify what they mean when they say "as a de-trended (zero-mean) time series data set"? I'm a little uncertain if a linear trend is removed or if it's just the annual cycle that's removed. If no trend removal is done then I think that could have implications. e.g. Line 311 and 370: "In particular, the correlation across the whole Eurasian–Pacific sector of the Arctic has been more strongly negative since 2000". If there's a positive trend in the AO since 2000 (I'm not sure, but it looks like it), and a negative sea ice trend then that could explain the changing relationship between the AO and sea ice, but it might not be causal (e.g. global warming as a confounding variable).

For all data, the linear trend is removed. We have now stated this explicitly in the revised manuscript to avoid any confusion.

I think when comparing models to observations it would be useful to have a benchmark that accounts for natural variability. For example, it would be interesting to compute ARI and D for CanESM5 by taking an ensemble member as truth (or ideally looping through ensemble members taken as truth). This would give a better benchmark to compare ARI and D to, and could possibly be included on Figure 11. It could also be possible to do a kind of bootstrap where only a random subset of years is selected from observations, but I'm less sure how that would work. In general I think a bit more of an acknowledgement that some of the differences between networks could be due to sampling of internal variability e.g. Line 235: "MIROC-ES2L model produces the most dissimilar network structure relative to ERA5", Line 264: "suggests that the models show large disagreement on the degree of connectivity".

Thank for you for this great suggestion. A similar point was also raised by reviewer 1, and we agree that some discussion on this needs to be included. To re-iterate our response to reviewer 1: In our case the 20 CanESM5 ensemble members actually comprise two groups of 10 members with different physics (group 1: i1p1f1, group2: i1p2f1), therefore while we can highlight the spread due to internal variability in ARI and D metrics across these members (for each group, physics 1 and physics 2), it will still likely be an under-estimate due to limited sampling. Nonetheless we have included some discussion and additional figures in the revised manuscript.

Towards the end of the paper an explanation of some of the mismatch between models and observations on how the winter AO influences summer sea ice, however the majority of the difference in pattern remains unexplained. I just wanted to comment that I hope the authors (or some other reader of this paper) do pursue this issue further using these complex networks in future work, as the results would be extremely interesting.

Thank you for this comment. We agree that further investigation is required to properly investigate this teleconnection, and all the intermediate processes that are affected by the winter AO, that subsequently drive the ultimate response of summer sea ice. We hope that this paper at least lays some initial groundwork for any future investigation, to be continued by either ourselves or others in the community at a later date.

**Technical corrections**

258: minor nitpick - fraction, not percentage
We have edited this to "fraction".

389: I would guess this should be models' instead of model's
Yes, thank you!

Figures 13, 15: Should the covariance have some units when comparing the AO to sea ice concentration? Something like % per standard deviation in AO index? I'm not really sure how to interpret the 10^8, particularly as that doesn't really fit in with what I would think units might be.
In the revised manuscript we have added units to all plots which show covariance.

Figure 14: I'm not really sure what the legend is saying here.
Apologies, this appears to have been a bug in the file upload. The revised version should contain the correct legend.

---

## Referee Report (RR1)

**Review of the revised manuscript „Network connectivity between the winter Arctic Oscillation and summer sea ice in CMIP6 models and observations" by Gregory et al.**

I thank the authors for their thorough revision of the manuscript. They addressed all my comments in a satisfactory manner. I especially appreciate their effort in adding a discussion on the role of internal variability to the results. Overall, this is a very interesting and well-prepared study and I recommend the revised manuscript for publication after consideration of a few additional minor/technical comments (line numbers refer to the version with tracked changes).

- L250 and in the captions of Figure 4, 6, 10, 15: The ensemble members in an initial-condition large ensemble have the same physics and forcing but slightly different/perturbed initial conditions (see e.g. Deser et al., 2020: https://www.nature.com/articles/s41558-020-0731-2). I suggest changing "which contain the same initial conditions, physics, and forcing" to "which contain the same physics and forcing but slightly perturbed initial conditions" or similar.

- Thanks for adding the discussion on internal variability based on the CanESM ensemble. What I would find even more interesting to discuss than the comparison of the two sub-ensembles with different model physics (p1/p2) is a comparison of the CanESM ensemble (intra-model spread) to the entire CMIP6 multi-model ensemble (inter-model spread), which can give an indication of how much of the spread is due to internal variability and how much is due to different model physics. Particularly for the AO – sea ice teleconnection (Fig. 13), the spread in D seems to be as large in the CanESM model as within the CMIP6 models. But I also understand that this aspect cannot be discussed in all details here and also accept the current version.

- I find Figs. 6, 10, 15 very interesting but could also see them going into the supplementary material to keep the main paper concise (up to the authors).

- Thanks for adding the units to the covariance. It took me a moment to verify because I didn't remember that SLP is on a lat-lon grid while SIC is on a polar grid. Maybe you can add that information to the text in L146, e.g.: "[...] for a regular latitude-longitude grid (sea-level pressure data), where theta is the latitude of grid cell p, or simply $w_p = d_p$ for a polar stereographic area grid (sea-ice concentration data), where $d_p$ is the area in $km^2$ of grid cell p [...]".

  Also, it is not quite clear to me why you don't square-root the area weights for the SIC anomaly time series in the computation of the AO – sea ice covariance (L340-343). It seems a bit inconsistent to me, but maybe I'm missing something here. I also think that keeping [Pa $km^2$] (or [Pa km]) as the unit might be more intuitive than Newton.